# Single-cell transcriptomics reveal transcriptional programs underlying male and female cell fate during *Plasmodium falciparum* gametocytogenesis

Mubasher Mohammed[1] ✉, Alexis Dziedziech[1,2], Diego Macedo[1], Frederik Huppertz[1], Ylva Veith [1], Zoé Postel [3], Elena Christ[1], Richard Scheytt [1], Tanja Slotte [3], Johan Henriksson [4] & Johan Ankarklev [1,5] ✉

The *Plasmodium falciparum* life cycle includes obligate transition between a human and mosquito host. Gametocytes are responsible for transmission from the human to the mosquito vector where gamete fusion followed by meiosis occurs. To elucidate how male and female gametocytes differentiate in the absence of sex chromosomes, we perform FACS-based cell enrichment of a *P. falciparum* gametocyte reporter line followed by single-cell RNA-seq. In our analyses we define the transcriptional programs and predict candidate driver genes underlying male and female development, including genes from the ApiAP2 family of transcription factors. A motif-driven, gene regulatory network analysis indicates that AP2-G5 specifically modulates male development. Additionally, genes linked to the inner membrane complex, involved in morphological changes, are uniquely expressed in the female lineage. The transcriptional programs of male and female development detailed herein allow for further exploration of the evolution of sex in eukaryotes and provide targets for future development of transmission blocking therapies.

Eukaryotic microorganisms are known to reproduce asexually or sexually, where asexual reproduction generates rapid means for population expansion through a largely homogeneous population. Sexual reproduction, which typically involves the formation of gametes that fuse to create a zygote, leads to advantages in genetic variation held in the population, given the recombination events which beget genetic diversity[1]. In *Plasmodium spp.*, the etiological agent of malaria, both asexual and sexual reproduction occur. Asexual replication occurs in the infected red blood cells (iRBCs) and may lead to

parasite numbers in the billions per infected host[2]. However, a small minority of the asexually replicating parasites differentiate to initiate sexual stage development. *P. falciparum* male and female precursor gametes, referred to as gametocytes, develop over 10–12 days, including five distinct morphological stages (stages I–V), which is significantly longer than that of other *Plasmodium spp.*[3]. The gametocyte stage is the only life cycle stage that can be effectively transmitted to the *Anopheles* mosquito vector where gamete fusion followed by meiosis occurs[4]. Recent advances in malaria control have bolstered the

[1]Department of Molecular Biosciences, The Wenner-Gren Institute, Stockholm University, Stockholm, Sweden. [2]Department of Global Health, Institut Pasteur, 25-28 Rue du Docteur Roux, Paris, France. [3]Department of Ecology, Environment and Plant Science, Stockholm University, Stockholm, Sweden. [4]Laboratory for Molecular Infection Medicine Sweden (MIMS), Department of Molecular Biology, Umeå Centre for Microbial Research (UCMR), Integrated Science Lab, Umeå University, Umeå, Sweden. [5]Microbial Single Cell Genomics Facility, SciLifeLab, Biomedical Center (BMC) Uppsala University, Uppsala, Sweden. ✉e-mail: mubasher.mohammed@ki.se; johan.ankarklev@su.se

interest in malaria intervention strategies linked to the reduction of transmission, including vaccines that target sexual, sporogonic, and/or mosquito-stage antigens in order to interrupt malaria transmission (SSM-VIMT)[5]. Since gametocytes do not replicate, a fitness trade-off between growth and reproduction is determined through adaptive phenotypic plasticity (APP), which allows the parasite to adapt to the host organism[6]. The gametocyte conversion rate has been suggested to be determined by host factors, although very few have been determined with the exception of host-derived lysophosphatidylcholine (LysoPC)[7]. LysoPC has been shown to repress *P. falciparum* sexual differentiation. It has also been shown that population density cues, including microvesicles released from iRBCs, can positively influence parasite sexual-stage commitment[6,8]. Gametocyte induction is further dependent on pH, temperature, stress, and likely other, hitherto unknown factors[9–11].

It was discovered by Kafsack and colleagues that the ApiAP2-G transcription factor from the Apicomplexan Apetala 2 gene family is essential for sexual stage commitment and differentiation[12]. Further, it has been suggested that histone modifications are involved in creating a genetic landscape for alternative transcriptional programs, starting as early as the asexual ring stage[13]. For asexual-stage parasites to commit to sexual-stage development, the epigenetically controlled transcription factor, AP2-G, has to become activated, leading to the specific expression of gametocyte-specific genes via the release of HP1-mediated heterochromatin. This in turn is mediated by the perinuclear protein, GDV1 (gametocyte development 1), and the silencing of asexual genes in early gametocytes via H3K36me2 and H3K36me3[12,14,15]. After sexual commitment, *ap2-g* expression has been shown to be repressed in stage II gametocytes due to its association with H3K36me2&3 silencing along with *hp1*[14]. In addition, parasites with disrupted *ap2-g2* expression were unable to mature beyond stage III[16]. In *P. berghei* it has been shown that female gametocyte development requires the involvement of *Pb*AP2-FG, and that *Pb*AP2-FG together with *Pb*AP2-R2 function as a repressor complex essential for *P. berghei* female maturation[17]. However, the involvement of the *Pb*AP2-FG ortholog in *P. falciparum* or other *Pf*AP2 transcription factors in regulating male or female-specific transcriptional programs remains unclear. Given that the transcription factors of the ApiAP2 family have been shown to be important regulators of differentiation and development throughout the life cycle in *Plasmodium spp.*, further exploration of their involvement during *P. falciparum* gametocytogenesis is necessary.

Two recent studies have characterized the influence of specific genes on the male and female lineage during gametocytogenesis. Gomes and colleagues uncovered a male sex-determining factor (md1) in *P. falciparum*[18]. In parallel, Russell et al. screened and identified genes required for sex-specific gene expression programs in the rodent infecting malaria parasite, *P. berghei*[19]. Both studies pointed to a possible connection of RNA-dependent regulation implicating different variants of RNA species in sex determination and subsequent lineage development. This has been confirmed by the discovery of md1 and the role of splicing kinetics in the bifurcation of the male lineage in *P. falciparum* gametocytes. Regulation of alternative splicing has been shown to be integral for the proliferation of the asexual blood stage[20], however, the downstream role during *P. falciparum* sexual cell fate is yet to be uncovered.

Due to the haploid nature of blood-stage parasites, *P. falciparum* sex determination does not involve inherited mating type loci or sex chromosomes. Instead, it has been suggested that sexual cell fate is determined through tightly coordinated epigenetic regulation, such as the involvement of histone modifications and alternative splicing[21,22]. Sexual stage commitment occurs as early as during the ring stage, but it is not known when the sexual cell fate decision occurs that leads to divergent transcriptional programs and results in mature male and female gametocytes. The mechanisms involved in determining

different sexes are known to evolve rapidly[23] and Apicomplexa lack genetic homologs for sex determination found in other eukaryotes[19]. Thus, high-resolution transcriptome mapping at the single-cell level would be an effective approach to delineate the timing of, and genes involved in, driving the differential *P. falciparum* gametocyte transcriptional programs. This would also enable lineage-specific tracing of the involvement of the *Pf*ApiAP2 family of transcription factors during the development of male and female *P. falciparum* gametocytes.

We previously generated fluorescence-expressing transgenic lines, using *P. falciparum* genetic markers known to be expressed throughout gametocyte development[24]. In this study we utilized one of these lines, designed with the gametocyte-specific Peg4 promoter, driving tdTomato fluorescence, in order to enrich parasites through their development from gametocyte rings to mature male and female gametocytes, by the use of FACS. Subsequently, we performed single-cell transcriptional profiling to uncover the diverging transcriptional programs underlying *P. falciparum* gametocytogenesis. Using computational analyses, we identified key genes, which are predicted to drive the maturation of male and female gametocytes by investigating the role of splicing kinetics throughout development. Our data indicates differences in male and female lineage-specific expression of a majority of genes known to be involved in the formation of the inner membrane complex (IMC), indicating a more prominent role in the development of the female lineage. Further, we delineated the timing of expression of specific *P. falciparum* ApiAP2 transcription factors and predicted their involvement in the male or female transcriptional programs. Finally, we used available DNA motif position weight matrices[25] to generate a gene regulatory network of predicted target genes regulated by specific ApiAP2 transcription factors. In total, we anticipate that our study provides the malaria community with an important resource to further our understanding of *P. falciparum* male and female gametocyte development, as well as additional data for intervention strategies targeting malaria transmission.

## Results

### Unsupervised clustering of developing *P. falciparum* gametocytes reflects a dimorphic transcriptome

To characterize *P. falciparum* gametocytogenesis, we have generated single-cell transcriptomes covering the entire gametocyte developmental phase. For this, we used a transgenic *P. falciparum* strain, designed with the promoter of the gametocyte-specific *Peg4* gene driving tdTomato fluorescence, which enables stage-specific enrichment using fluorescence-based cell sorting. Thus, parasites were FAC-sorted prior to loading the enriched populations on a 10X Chromium Controller (Fig. 1A; Supplementary Fig. 1A). QC of the sequence data, including quality filtering, rendered a total of 4555 cells, with an average of 986 genes expressed per single cell (Supplementary Fig. 1B, C). The number of genes and transcripts detected is provided in Supplementary Data 1. We performed dimensionality reduction and unsupervised clustering implemented in the Scanpy pipeline[26], followed by cell-type correlation analysis. The results were projected using Uniform Manifold Approximation and Projection (UMAP) analysis (Fig. 1B; Supplementary Fig. 1E), including ten unique gametocyte cell clusters (C0 to C9). The Wilcoxon rank test was used to distinguish genes that represent specific cluster markers. We evaluated the top 2 most differentially expressed genes across all clusters and identified them as associated with different developmental stages (Fig. 1C; Supplementary Data 2). P48/45 (PF3D7_1346700) and PMIX (PF3D7_1430200) are markers for *Plasmodium* gamete fertility, and invasion and development, respectively[27,28]. Both genes were found highly expressed in C2, with P48/45 showing a significantly higher log fold change difference in expression in C5 (Wilcoxon rank test, adj. *p* value < 0.05). P48/45 has previously been linked to fertilization in male gametocytes[29], proposing its association with the male gametocyte cell population in C5. While PMIX expression is upregulated

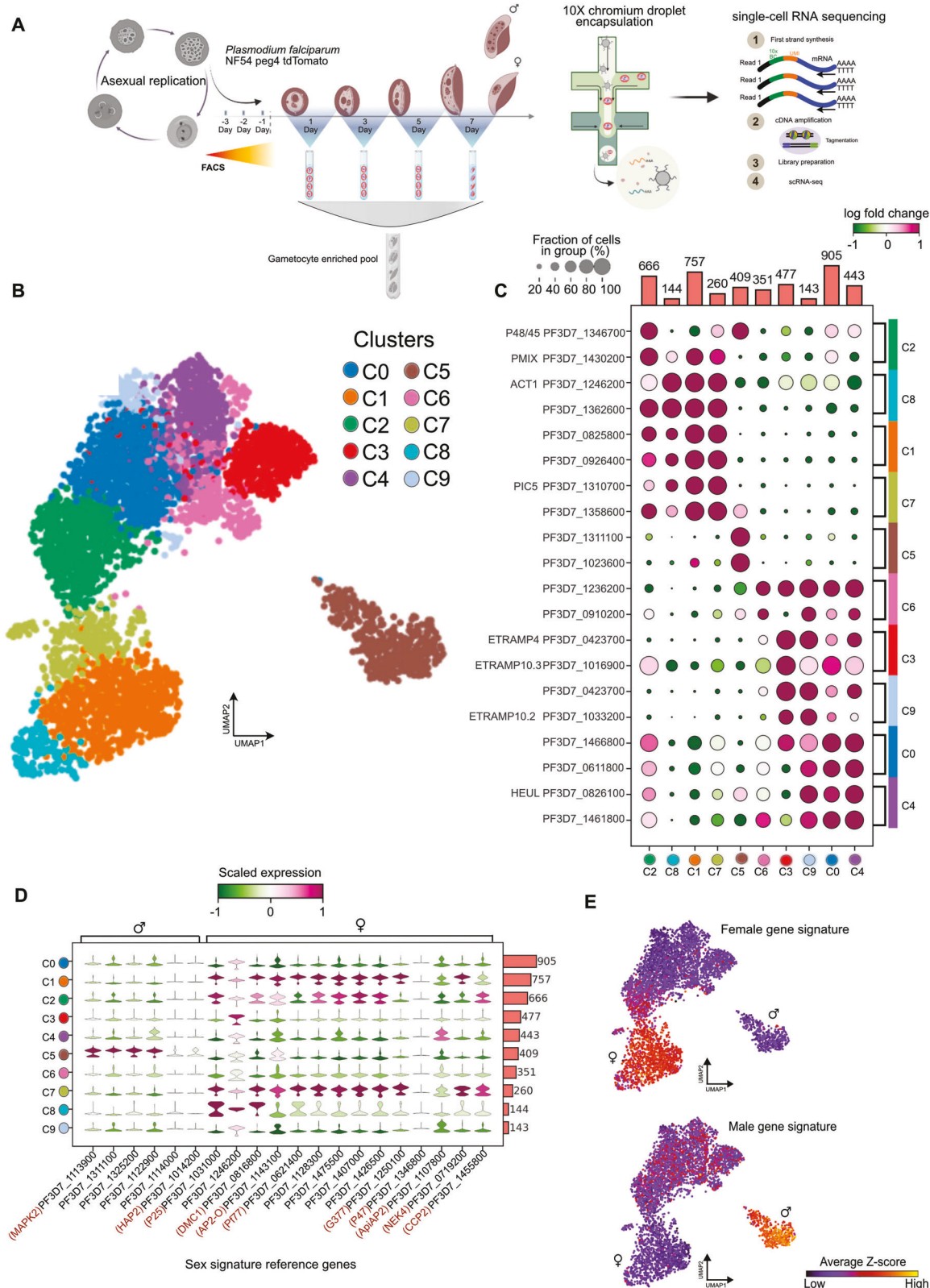

approximately one log fold in the C1 and C7 clusters, with undetected expression in C5, indicating its association with the female gametocyte cell population (Fig. 1C).

Previous genome-wide transcriptome studies demonstrated the expression of the Early-Transcribed Membrane Proteins (ETRAMP) gene family during the early stages of sexual development[30]. Three members of this gene family; *ETRAMP10.3, ETRAMP4*, and *ETRAMP10.2*, were expressed among clusters; C0, C3, C4, and C9 (Fig. 1C). ETRAMPs are localized mainly on the parasite-host-cell interface and play a key role in nutrient uptake assisting the parasite's survival and development[31].

To determine if any of the clusters were associated with previously reported canonical markers of mature female and male gametocyte genes, we checked for canonical sex markers across the

**Fig. 1 | Single-cell transcriptomics targeting the *P. falciparum* sexual stages.**
**A** Schematic outline, illustrating the experimental approaches used to characterize transcriptional changes during *P. falciparum* male and female gametocyte development, using the *P. falciparum* gametocyte-producing NF54 peg4-tdTomato transgenic cell line. The schematic highlights FACS-based parasite population enrichment followed by 10X 3' single-cell RNA-seq (10X Genomics). Panel **A** was created with BioRender.com released under a Creative Commons Attribution-NonCommercial-NoDerivs 4.0 International license (https://creativecommons.org/licenses/by-nc-nd/4.0/deed.en). **B** Uniform Manifold Approximation and Projection (UMAP) dimensionality reduction and unsupervised Louvain clustering of 4555 single-cell gametocyte transcriptomes. Two independent single-cell libraries (PF01 and PF02) were pooled for combined analysis, PF01 ($n = 2140$ cells) and PF02 ($n = 2415$ cells). **C** Dot plot visualizes the top two differentially expressed genes among heterogeneous cell type populations (clusters) using the Wilcoxon rank-sum test (two-tailed) implementing the Scanpy function "rank_genes_groups" with

standard settings and a Bonferroni adjusted *p*-value < 0.05. Color scale indicates the log fold change (FC) of the expression with green indicating relative negative expression and dark red indicating relative positive expression. Dot sizes represent the proportion of cells within a cluster that express a certain gene, only genes with minimum log fold change ≥1.0 and Bonferroni adjusted *p*-value < 0.05 are shown. A total number of 4555 cells were analyzed from two sample libraries. The number of cells per cluster are added as a bar graph to each heatmap. **D** Stacked violin plots depicting representative sex-specific markers scaled expression from −1 (dark green) to 1 (dark red) across clusters. **E** Sex-specific reference gene signature average Z-score projected across the UMAP, enrichment of previously identified female (top) and male (bottom) sex-specific markers (dark blue indicates a low average Z-score and bright yellow a high average Z-score). The gene signature score is calculated using the Scanpy function "tl.score_genes". The male and female gametocyte marker genes used for this analysis have been implemented from the study by Walzer and colleagues[32].

clusters[32]. Similar to the correlation analysis in Supplementary Fig. 1E, clusters C0, C3, C6, and C9 had similar downregulated expression profiles of female marker genes whereas C1, C2, and C7, on the contrary, showed upregulated expression profiles of the same panel of marker genes (Fig. 1D). In contrast, C5 is characterized by a distinct signature of upregulated male canonical marker genes defining the mature male gametocyte cell population in C5 (Supplementary Fig. 1I). Hence, the high expression of these sex-associated genes plays a major role in separating the early and late gametocyte transcriptome signature as observed in Supplementary Fig. 1E. The estimated average Z-score of reference signature markers for both gametocyte sexes was projected on a UMAP showing distinct patterns identifying the mature male and female gametocyte transcriptome (Fig. 1E).

## Gametocyte cell fate annotation using discriminative dimensionality reduction

To capture differential gene expression (DGE) programs associated with sexual cell fate, we employed discriminative dimensionality reduction of pseudotime (DDRTree). The reconstructed trajectory towards male or female lineage consisted of two main terminal branches forming a bifurcation point preceded by a pre-branch structure in both the clusters and combined analyses reflecting three main cell states (Fig. 2A, B, respectively; Supplementary Fig. 2A). To gain insight into the biological variance defined by the observed cell states, we then mapped the cell states to the PCA included in Supplementary Fig. 1H and visualized the cells in reduced dimensional space. The variation of RNA profiles of the assigned cell states is represented by the first two components covering the bifurcation point of differential female and male transcriptional signatures from the common progenitor cells (Fig. 2D; Supplementary Fig. 1C). To further disclose the distinct genetic signature of the cell state heterogeneity, we performed a differential gene expression analysis using the Wilcoxon rank test. Among the top 50 upregulated genes across the cell states (Supplementary Data 3), we found previously reported markers of female gametocytes. This includes; the plasma membrane-associated protein P25, DMC1, a meiotic recombination protein, and the female gametocyte-specific transcription factor AP2-O, described to likely play a role in the progress of sex differentiation[32,33]. This branch was annotated as the 'Female lineage' (Fig. 2C, E; Supplementary Data 3). In our modeled trajectory, cells differentiated along the second branch were characterized by expression of meiosis-specific nuclear structural protein 1 (PF3D7_1311100), putative lactate dehydrogenase (PF3D7_1325200), and dynein heavy chain, putative (PF3D7_1122900). These genes were previously shown to be expressed in mature male gametocytes and supported the annotation of this branch as the 'Male lineage'[32] (Fig. 2C; Supplementary Data 3). The pre-branch was associated with distinct gene signatures reported to be expressed during early gametocytogenesis, including, *EXP1* (PF3D7_1121600) important in nutrient uptake, *P113* (PF3D7_1420700), known to encode a membrane-

anchored protein, *ETRAMP4* (PF3D7_0423700), playing a role in nutrient uptake, and *HDP1* (PF3D7_1466200), a transcription factor, which were among the top 50 ranked upregulated genes (Supplementary Data 3). Thus, we annotated this pre-branch as 'Early gametocyte'. After the annotation of the trajectory inference branches, we tested for sex-specific gene signature scores and observed agreement between lineage cell scoring and our branch annotation based on differential gene expression testing across pseudotime branches. The male lineage shows the highest score for canonical male signature genes (Supplementary Fig. 1D, top panel), and the female gametocyte branch lineage showed the highest score for canonical female signature genes (Supplementary Fig. 1D, bottom panel). These results support the cell lineage annotation generated from inferring the pseudotime trajectory using DDRTree-based classification.

To assess the distribution of our single-cell data to previously defined post-commitment sexual stages, we deconvoluted the existing bulk gametocyte transcriptome from van Biljon and colleagues[34] to map the gametocyte developmental stages (including, I Day 3 to IV Day 8) to our scRNA-seq data. We used a deconvolution linear regression model to calculate the cell type proportion using the shared expression profile between the bulk RNA-seq data and our single-cell data. We fit the regression model to transfer the gametocyte stage and time point labels into our single-cell data (Supplementary Fig. 2A). We then projected the identified labels from the bulk gametocyte transcriptome into our previously performed PCA and pseudotime trajectory (Supplementary Fig. 2B) The resulting mapping indicated that the bulk labels align with our pseudotime data for these stages (Supplementary Fig. 2B). In addition, we analyzed the correlation between our scRNA-seq data with the bulk RNA-seq data from the van Biljon et al. study, our pseudo-bulk alignment showed the relative clustering between our single-cell data and the bulk RNA-seq data (Supplementary Fig. 2C). We further wanted to investigate the cell type sex allocations over the course of the pseudotime-bulk alignment. The data showed that the male lineage differentiation still occurred at or prior to what is referred to as day 4—stage I to II in the van Biljon study with the majority of male lineage cells appearing to express a high number of mature male transcripts on day 6, as it is referred to in the van Biljon study (Supplementary Fig. 2C).

In addition, we used the single-cell transcriptome dataset from Gomes et al.[18] (Supplementary Fig. 3A) and integrated it with our single-cell transcriptome dataset using Scanorama v1.7.4 to obtain a scalable gene expression profile across the two datasets. PCA analysis enabled us to observe a clear alignment between the two datasets including a developmental trajectory bifurcation when using the Gomes et al. single-cell data from the WT parasite line (Supplementary Fig. 3B). Further, a lack of male gametocytes was observed when comparing our dataset with the Gomes et al. single-cell data from the *md1* knockout parasite line (Supplementary Fig. 3C). Our data indicates the following ratio of cell types: progenitor cells = 42.6%, female

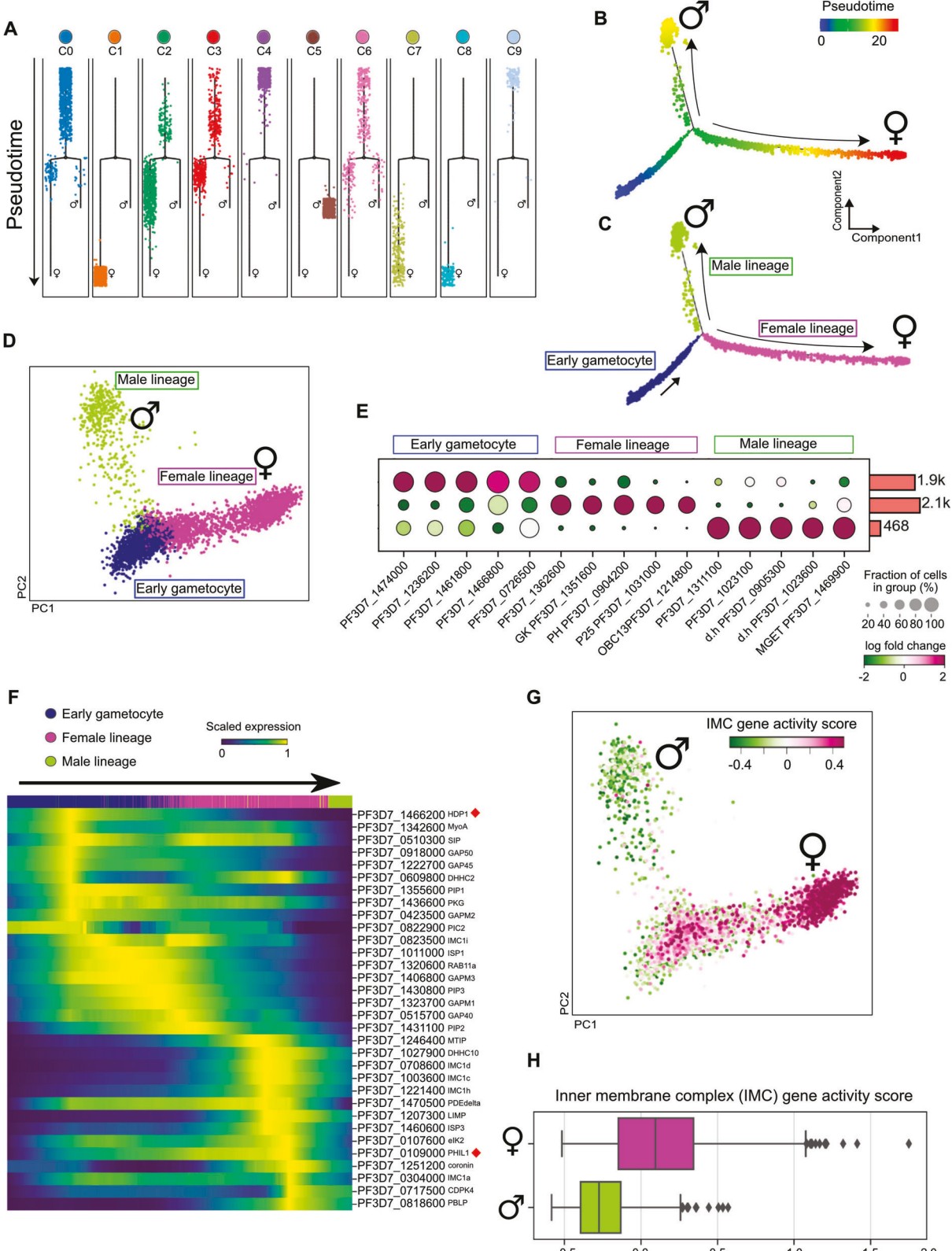

lineage = 47.2% and male lineage = 10.3%, in comparison with the Gomes et al. data, which indicates the following ratio of cell types: progenitor cells = 36%, female lineage = 58.7% and male lineage 5.3% (Supplementary Fig. 3D).

Apicomplexan parasites possess a unique set of organelles called the apical complex, which is essential for host cell adhesion, invasion and the establishment of the parasitophorous vacuole[35,36]. A previous study has shown that a family of genes, stemming from the apical complex, including the inner membrane complex (IMC)-associated genes, are essential for gametocyte morphology and are controlled by the transcription factor, HDP1 (PF3D7_1466200)[37]. HDP1 binds two sites upstream of the transcription start site (TSS) of a subset of IMC target genes to expand the inner membrane complex that gives *P. falciparum* gametocytes their distinctive shape.

**Fig. 2 | Discriminative decision tree classification captures the early transcriptional switch of male and female gametocyte differentiation. A** Cell trajectory plot visualizing the transcriptional progression of cells, from each population cluster, over the pseudotime developmental trajectory to identify the bifurcation point of the male and female gametocyte lineages. The trajectory length (black line) represents the pseudotime units with cluster assignment generated using DDRTree. **B** Representation of a reconstructed cell developmental trajectory depicting pseudotime units generated from DDRTree for the simulated two-branch points (n = 4555 cells). **C** The cell developmental trajectory of each branch was validated with reference genes for early, male, and female gametocytes. **D** PCA plot overlaid with cell class (or state) annotations generated in the DDRTree classification. Transcriptional variance of cells along PC1 and PC2 reflects transcriptome signature variation between the male (green) and female (pink) gametocyte lineages, stemming from the relatively homogeneous single-cell transcriptomes of the gametocyte progenitor cells (purple). **E** Dot plot showing the top 5 differentially upregulated genes across the pseudotime trajectory cell class annotation, using the Wilcoxon rank test with corrected Bonferroni adjusted p-value < 0.05. The color

scale depicts log fold changes in expression from −2 (dark green) to 2 (dark red). The dot sizes depict the proportion of cells from each cluster that express a certain gene. A total of 4555 cells from two sample libraries were included in this analysis. **F** Pseudotime heatmap of scaled expression of genes selected for inner membrane complex (IHC) components. Genes are ordered based on their individual corresponding pseudotime units in addition to the HDP1 transcription factor. The color scale indicates low (dark blue) to high (yellow) levels of gene expression.
**G** Projection of IMC gene expression activity across the PCA, with low (dark green) to high (dark pink) levels of gene expression. **H** Box plot depicting the gene activity score for IHC genes and/or genes involved in membrane formation and shaping of gametocytes morphology for the female (upper) and male (lower) lineages. Box plots are centered based on the average expression of a set of IMCs genes subtracted from randomly sampled male (n = 468 cells) and female (n = 2100 cells) lineage gene pools. The box represents the first to third quartiles and the whiskers extend with a maximum of 0.5X the interquartile range, points beyond this range are plotted as outliers.

Our analysis of differentially regulated genes across trajectory branches indicated that *HDP1* is upregulated in the early gametocyte cell state (Supplementary Fig. 1J). To uncover the temporal dynamics of *HDP1* expression in connection to a set of IMC genes, we used the pseudotime ordering of genes across the identified cell states (early, female, and male gametocyte lineages). Our results indicate that the expression of the IMC genes is preceded by HDP1 (Fig. 2F). In addition, we observed that genes, including *MyoA, GAP50, GAP45*, and *PIP1* are expressed at their highest levels during the early gametocyte cell state. While, *IMC1d, IMC1h, IMC1c, PhIL1*, and *IMC1a* expression peaked during the later stages of the female gametocyte cell state, with relatively low to no expression of IMC genes within the male gametocyte lineage (Fig. 2F).

Previous reports have shown the localization of basal bodies near the apical pole during micro-gametogenesis in *P. falciparum*[38], but the role of IMC and IMC-associated protein-encoding genes specifically linked to gametocyte male or female lineage development has previously not been described. To investigate differences in IMC gene expression between male and female gametocyte lineages, we calculated the average z-score of these genes and visualized their expression activity score on PCA (Fig. 2G). The IMC gene activity score suggests that the female gametocyte lineage displays a higher expression activity when compared to the male lineage (Fig. 2F–H).

## Splicing kinetics define lineage-specific genes during *P. falciparum* gametocytogenesis
To delineate the involvement of specific genes in driving the progression of male or female maturation, we employed RNA velocity[39] which ultimately compares the number of reads originating from pre-mRNA (unspliced) and mature mRNA (spliced). The unspliced mRNA count reads were generated from the initial mapping of the *P. falciparum* genome and the RNA velocity vector field overlaid across identified clusters on UMAP (Fig. 3A). RNA velocity predicted two independent trajectories, initiated from clusters (C0, C3, C4, and C9), with terminal points in C5 and C1 (Fig. 3A), previously annotated as mature male and female gametocyte cell populations. These findings were further supported by cell-to-cell similarity trajectory mapping as in Supplementary Fig. 4A, indicating that our cell class annotation was reproducible in two different cell trajectory analyses. To further validate our cell RNA velocity-based annotation, we compared the ratio of spliced and unspliced mRNA for a male marker, dynein heavy chain, putative (PF3D7_1122900)[40], and a female marker, *DMC1* (PF3D7_0816800) a meiotic recombination protein, essential in the meiotic division and recombination[34,41]. For the male marker, we observed a ratio of nascent RNA reads that were elevated in early gametocyte cell population clusters, barely detectable in C1, the mature female cell population, and mainly detected in C5, the mature

male gametocyte population (Fig. 3B). Conversely, *DMC1* showed elevated expression in the mature female gametocyte cell population[32], while its unspliced mRNA were distributed over the course of the early gametocyte cell population clusters (Fig. 3C).

In addition, we ranked the cluster-specific genes that presented differential velocity expression in C1 and C5, by measuring the ratio between pre-mature mRNA and mature mRNA for the top 5 velocity genes, allowing the identification of the mRNA splicing kinetics of several genes in the mature male and female populations (Fig. 3D, E). For C1, annotated as mature female gametocytes, we found genes including; *TRF1* (PF3D7_0924800), *PSOP13* (PF3D7_0518800), *41-3 protein* (PF3D7_1207700), PF3D7_1005200, and *LIMP* (PF3D7_1207300), among the top-ranked genes showing a high number of unspliced mRNA (Fig. 3D). TRF1 is suggested to act as a negative regulator of telomere length[42], and is constantly expressed through the life cycle of the parasite with a spike in gametocytes, rising along the cell maturation[43,44]. LIMP and PSOP13 have been shown to be transcribed in female gametocytes but not translated until the ookinete stage, their association with a high number of unspliced mRNA suggests the early transcripts production of these genes during the female lineage development, while the translation to proteins is essential for ookinete gliding motility and invasion[45,46]. These findings indicate the presence of pre-mRNA maternal transcripts with female gametocyte intrinsic features and are captured by unspliced mRNA in the early gametocyte population. In C5, annotated as mature male gametocytes, we observed a significantly elevated expression of mature mRNAs coding for putative lactate dehydrogenase, *pLDH* (PF3D7_1325200), a canonical male gametocyte marker[45]. While this known annotated male gene was observed, several other top-ranked genes with a high number of unspliced mRNA remain unannotated (Fig. 3E), highlighting the necessity of further functional characterization of genes involved in male gametocyte maturation.

## Identification of putative lineage-specific driver genes using RNA Velocity
Distinct gametocyte lineages are typically governed by different variations in gene activity programs and exhibit different splicing kinetics. This gives rise to genes displaying an increased correlation to lineage-specific gene expression programs. To identify potential lineage-specific driver genes, we ran a generalized additive model (GAM) using a set of canonical mature male and female gametocyte markers to model expression trends towards either C1 (mature female) or C5 (mature male) (Fig. 4A; Supplementary Fig. 4A, B). Based on gene expression correlation values (q-value < 0.01), we identified and ranked lineage-specific putative driver genes holding a criteria of high expression correlation levels in cells that are biased towards the fate of either mature female or male gametocytes (Fig. 4B; Supplementary

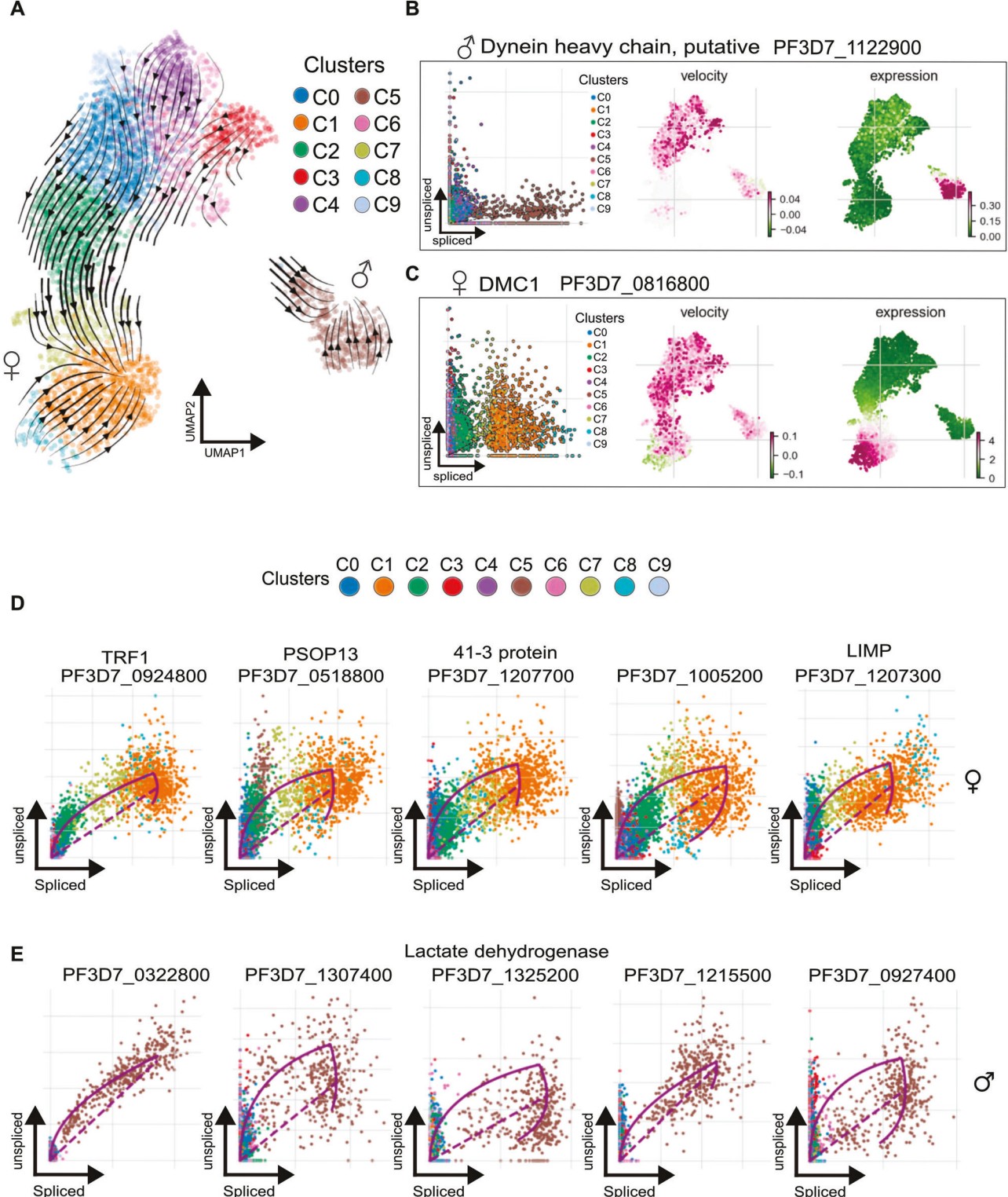

**Fig. 3 | Implementation of RNA velocity to delineate cell fate through splicing kinetics. A** Modeling of the transcriptional dynamics by projecting the RNA velocity data across the UMAP. The colors correspond to the UMAP cluster identities in Fig. 1B. **B**, **C** Validation of splicing kinetics of two known male, PF3D7_1122900 (**B**) and female, PF3D7_0816800 (**C**) sex-specific markers (left panels). Visualization of unspliced mRNA transcripts (middle column) in comparison with the gene expression trends (right panel) projected across the UMAP. The color scale depicts the velocity ratio (−1 to 1) or expression level (0 to 4) with the minimum values in dark green and maximum values in dark pink. **D**, **E** Scatter plots visualizing the top 5 ranked velocity genes (differential velocity expression t-test, threshold of minimum correlation set to 0.3), in UMAP cluster C1 (**D**, mature female gametocytes) and UMAP cluster C5 (**E**, mature male gametocytes).

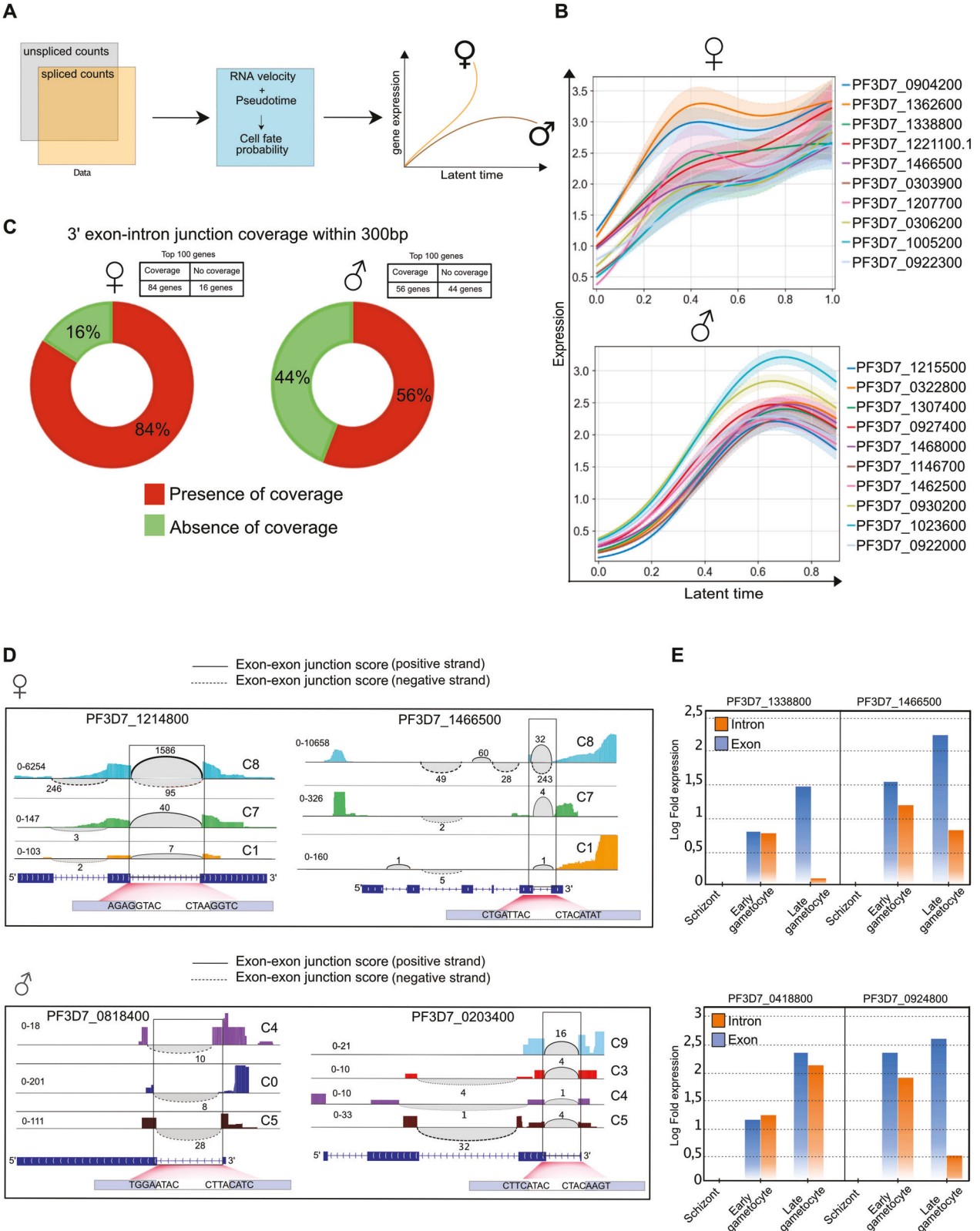

Data 4). In addition, we accounted for unspliced mRNA detection to validate the gene activity program over the lineage-tracing time course (Supplementary Fig. 4F, G). By correlating gene expression of the top 10 identified lineage-specific putative driver genes with clustering results, we observed gene expression gradients among female lineage cell population clusters (Supplementary Fig. 4C, E left panel). In contrast, the top 10 male lineage-specific putative driver genes showed high expression levels exclusively in C5 of the mature male population (Supplementary Fig. 4D and E right panel). Moreover, the distribution of the unspliced mRNA among the top 5 listed putative female lineage driver genes indicated increased levels of detected unspliced mRNA among early gametocyte progenitors towards the female lineage, however, we did not detect unspliced mRNA within the male gametocyte lineage (Supplementary Fig. 4F). The unspliced mRNA among

**Fig. 4 | Gametocyte cell fate determination using RNA velocity and cell-cell similarity to infer lineage-specific putative driver genes. A** Conceptual outline of the computational strategy, including the combination of RNA velocity and pseudotime to infer cell fate probability and predict lineage-specific putative driver genes. **B** Visualization of the continuous expression trends of the top 10 putative driver genes by modeling their expression correlation to either male (C5) or female (C1) gametocyte terminal states. Expression trends of the top 10 female putative driver genes, plotted against the latent time (top). Expression trends of the top 10 putative male driver genes, plotted against the latent time (bottom), line colors correspond to the respective color linked to each accession number. The ribbons represent the fate probabilities of gene expression with a 95% confidence interval, and margins of error of 0.015 are shown. **C** Pie charts depicting the percentages of lineage-specific putative driver genes, including (red) or lacking (green) the 3′ exon-

intron within 300 bp sequence coverage, estimated using the IGV software. **D** Sashimi plots visualize the splicing junction coverage detected at the 3′ exon-intron regions to validate the RNA velocity estimates. The plots include two putative female driver genes (top panel), and two putative male driver genes (bottom panel). The first exon-exon junctions from the 3′-end of the genes are highlighted (black box) and the colored coverage tracks correspond to identified Louvain clusters obtained through unsupervised clustering. **E** rt-qPCR analysis of synchronized bulk parasite populations used to validate the abundance of immature transcripts of putative driver genes between early and late gametocytes, where asexual stage schizonts are used as baseline control. Log fold expression values of four representative genes are measured between the three samples, using primer sets that include one reverse primer and two forward primers placed either inside the respective intron or in the subsequent exon, for each gene.

top-listed male putative driver genes were detected among early gametocyte progenitors towards the male lineage (Supplementary Fig. 4G).

To investigate potential limitations in the 3′-end RNA sequencing coverage, we used the Integrative Genomic Viewer (IGV) software to investigate sequence coverage of the putative lineage-specific driver genes. From the total number of identified genes in the female lineage, we found that 84% of the genes were covered with at least a few bases within 300 bp on the first 3′-end exon-intron junction, while 16% did not match this selection criterion (Fig. 4C; Supplementary Fig. 5A, B). Among the identified male lineage genes, we found 56% of the total number of genes showing the targeted coverage, while 44% did not show the 3′-end exon-intron junction coverage (Fig. 4C; Supplementary Fig. 5C, D; Supplementary Data 4). Thus, we selected genes that were covered by mapped sequence nucleotides for further validation (Fig. 4C).

To validate differences in the abundance of spanning reads split across splicing junctions, we examined differences in the number of reads covering each splicing junction for two selected putative female and male driver genes (Fig. 4D). For the female genes, we observed a decreasing number of reads in splicing junction coverage towards the mature female gametocyte cell fate (C1), between clusters C8, C7, and C1 (Fig. 4D top panel). We found a similar pattern for one of the selected male genes, PF3D7_0203400, albeit there was a more limited number of reads available for the selected male genes in the sequencing data (Fig. 4D bottom panel). These differences in the abundance of unspliced mRNA signify differential isoform usage captured by splicing kinetics between early and late-developing gametocytes.

To experimentally validate intron usage during different stages of gametocytogenesis, we conducted rt-qPCR analysis of four female genes that showed splicing junction coverage among the female lineage putative driver genes, and compared them to asexual stage schizonts as a baseline (Fig. 4E; Supplementary Table 1). Our data indicates gradual intron splicing once early gametocytogenesis has initiated and transcript variants implicated in gametocyte sex determination have been synthesized.

We extended our analyses to explore the underlying biological processes across identified sex-specific putative driver genes. GO term enrichment analysis through the PlasmoDB portal for the top 50 listed genes, indicated GO terms implicated in "microtubule-based movement" (GO:0007018), "cilium movement" (GO:0003341), and "microtubule-based process" (GO:0007017), were over-represented by male putative driver genes (Supplementary Data 4). Further, we observed genes (PF3D7_0718000, PF3D7_0922000, and PF3D7_1146700) that were linked to the GO term for "microtubules-based movement" (GO:0007018). These genes have been reported to play an important role in gametocyte shape, driven by microtubules anchored to the inner membrane complex (IMC) and powering the motility of the flagellated male gametes[47,48]. Conversely, the top 50 listed female putative driver genes showed enriched GO terms such as, "cell gliding"

(GO:0071976), "protein folding" (GO:0006457), and "DNA recombinase assembly" (GO:0000730) (Supplementary Data 4).

We found *LIMP* (PF3D7_1207300) and *PM8* (PF3D7_1465700) among the top listed female putative driver genes that were connected to the GO term "gliding motility" and were reported to be translationally repressed and maintained with the other maternal female transcripts to be utilized in the next life cycle stages inside the mosquito vector[49].

## rt-qPCR to Investigate the timing of expression of the putative driver genes

To further investigate the timing of expression of the putative male and female driver genes, we performed rt-qPCR of synchronized gametocytes in bulk, then harvested at early, mid, and late timepoints of development. The analyses were carried out using a panel of primer pairs designed to target nine of the top putative male driver genes (MD) and nine of the top putative female driver genes (FD) (Supplementary Table 2). The data indicated a general trend of male and female genes being expressed based on their predicted lineage. We saw a trend of lower expression at the early time point of gametocyte development followed by an elevated level of expression in the mid to late timepoints for both putative male and female driver genes (Supplementary Fig. 6A through C). In contrast, MD4 showed a lower level of expression in gametocytes when compared to the control sample of asexually replicating parasites. No signal was detected in the rt-qPCR analysis for MD5, likely due to the technical difficulties in designing primers for some of these genes, thus MD5 was excluded from downstream analyses.

## Single-cell rt-qPCR to validate sex-specificity of the putative driver genes

To cross-validate the sex-specificity of the putative male and female driver genes, we used capillary-based micromanipulation to isolate individual gametocytes from early and late stages of development (Supplementary Fig. 6D) and deposited the single cells in smart-seq2 lysis buffer in individual microcentrifuge tubes. The smart-seq2 chemistry was applied[4], followed by thermal cycling over 24 cycles. The samples were subsequently analyzed using rt-qPCR, (primers listed in Supplementary Table 2) together with previously published male gametocyte markers (PF3D7_1477700 and PF3D7_1438800), female gametocyte markers (PF3D7_1447600 and PF3D7_1031000)[50] and primers for the 18S rRNA gene (PF3D7_0112300). Single gametocytes were filtered according to the following criteria, the inclusion of the expression of at least one of the known marker genes for male or female gametocytes, as well as the expression of at least three of the putative driver genes, resulting in a total of 31 gametocytes that passed the cutoff. The gene expression profiles of the expression pattern among these gametocytes indicated clear distinctions between male and female cells. There was a majority of female gametocytes ($n = 26$), in accordance with the expected 1:6 male-to-female ratio (Supplementary Fig. 6E).

Among the gametocytes that express the known male markers, the majority of cells show high expression levels of MD1, MD2, MD3, MD7, and MD8, whereas MD4, MD6, and MD9 only show expression in one or two cells. MD5 was undetected in the bulk samples and was therefore not included in the single-cell analyses. Gametocytes that express the known female markers, also showed high expression of FD2-FD8 in 15–24 out of the 26 cells. FD1 and FD9 were significantly less frequently expressed, including only three and five cells, respectively. It is noteworthy that MD7 showed high levels of expression in six of the parasites that otherwise showed a female pattern. In addition, FD4 showed high expression in four, FD5 in three and FD6 in two of the male gametocytes.

## Evolutionary conservation of the putative sex genes

To determine the importance and evolutionary conservation of the identified putative female and male driver genes, we compared the selective pressure acting on putative male, female, and nuclear genes, which served as the control. To do so, we estimated dN/dS, the ratio of the non-synonymous substitution rate (dN) to the synonymous substitution rate (dS) between *P. falciparum* and its close relative *P. gaboni*. We found that both male- and female-driver genes were relatively constrained as dN/dS was substantially lower than 1 (dN/dS = 0.2 vs 0.16 for female and male-driver genes, respectively)[51]. However, in comparison to the control set of nuclear genes, both putative male- and female-driver genes had significantly higher dN/dS, while male and female genes did not differ from one another. This was true when comparing the subset of genes that were selected as well as when comparing sex genes to all genes (Supplementary Fig. 7A; Kruskal–Wallis chi2 = 18.13, df = 2, $P = 0.00012$; Dunn post hoc test between female and male, Z = 0.726, $P$ = n.s.). We verified our results by ruling out potential artifacts due to effects of gene expression levels on dN/dS (Supplementary Fig. 7B). We found that our results were unlikely to be driven by differences in expression level between sex-driving and control genes (Kruskal–Wallis chi2 = 13.412, df = 2, $P < 0.01$; Dunn post hoc test between female and male, Z = −2.700, $P$ = n.s.). Likewise, our results were robust when statistically controlling for gene length (Supplementary Fig. 7C; Kruskal–Wallis chi2 = 20.119, df = 2, $P < 0.001$; Dunn post hoc test between female and male, Z = 0.951, $P = < 0.001$). This evolutionary genetic analysis supports that while all genes were constrained, selection on the identified sex genes differed from that of the control genes, potentially indicating a higher frequency of positive selection on putative male and female driver genes.

## Motif-driven gene regulatory networks predict AP2 target genes regulation along the male and female lineages

The bistable master regulator AP2-G controls the initiation of sexual commitment through binding and thereby initiating the expression of early gametocyte genes[12]. The expression of AP2-G decreases at the early stages of gametocytogenesis, while other members of the AP2 family are required for the activation of the downstream gametocyte developmental gene program[52]. To gain insight into the role played by these regulators during gametocytogenesis, we profiled the transcriptional dynamics of these ApiAP2 genes and inferred their lineage-specific expression trends towards either mature female or male gametocytes (Fig. 5A; Supplementary Fig. 8). We then refined the selection for genes whose expression best correlated with the probability of reaching the mature female (C1) or mature male (C5) state, as terminal clusters (Fig. 5A). We observed expression trends of AP2 genes that correlate well with the female gametocyte lineage, including *ap2-o* (PF3D7_1143100), *ap2-o2* (PF3D7_0516800), and *ap2-fg* (PF3D7_1317200) (Fig. 5A top panel), which are expressed in the C1, C2, C4, C7, and C9 clusters (Supplementary Fig. 8). Another hitherto unassigned ApiAP2 gene (PF3D7_1107800), was observed to be expressed in stage IV and V female gametocytes. These findings suggest a lineage-specific correlation of ApiAP2 expression trends from the early common sexual progenitor gametocytes throughout the female lineage progression (Fig. 5A, B).

Next, we assessed ApiAP2 expression trends correlating with the male gametocyte lineage (Fig. 5A bottom panel). We observed elevated expression trends for *ap2-lt* (PF3D7_0802100), *ap2-g5* (PF3D7_1139300), *ap2-g2* (PF3D7_1408200), and *ap2-tel/sp3* (PF3D7_0622900) with the male lineage (Fig. 5A, B). Apart from their elevated expression among early gametocyte clusters (C0, C4, and C9), these genes display an increased log fold change in the male gametocyte cluster (C5) (Supplementary Fig. 8).

Computational algorithms and publicly available datasets can be used to identify master regulators and potential target genes, rendering motif-driven gene regulatory networks (mGRN). We therefore, subsequently sought to identify putative transcription factors and regulatory genes in our data, by implementing mGRN analysis.

We used publicly available DNA-binding ApiAP2 position weight matrices generated from protein binding microarray (PBM) experiments[25] to build a *cis*-target database for *P. falciparum*. We then performed SCENIC (Single-Cell rEgulatory Network Inference and Clustering)[53] to infer co-expression modules between members of ApiAP2 transcription factors and target gene candidates to generate mGRN. Transcription factors were pruned based on the enrichment of the TF motif up to 1.5 kb upstream of the TSS of the potential target genes. Our analyses predicted a total of 8 enriched regulons divided into five "activating" regulons, including PF3D7_1107800, PF3D7_1222400, PF3D7_0420300, AP2-G5 (PF3D7_1139300) and PF3D7_1239200, and three "repressing" regulons, composed of AP2-G5 (PF3D7_1193300), AP2-G2 (PF3D7_1408200) and AP2-O4 (PF3D7_1350900) (Supplementary Fig. 9A, B). We then calculated the regulon enrichment score to identify regulons specific for early, female, and male gametocytes (Supplementary Fig. 9B). In addition, we projected these enrichment scores and their respective TF on a reduced dimension (PCA) for visualization (Supplementary Fig. 9A). These regulons are associated with a variable number of top interacting target genes that have been validated through manual detection of the motif sequence 1.5 kb upstream of the TSS using PlasmoDB (Supplementary Data 5). We then annotated and visualized the mGRNs into modules (Mod), including Mod A–E, based on interaction connectivity between the regulators and predicted downstream target genes using Cytoscape[54]. Mod A is composed of AP2-G5 (PF3D7_1139300) and PF3D7_0420300 as main regulators (Fig. 5C), both are expressed in early gametocytes and AP2-G5 shows elevated expression towards the mature male gametocytes (Fig. 5A, B). AP2-G5 has been described as crucial for both sexual commitment and development as it is implicated in the transcriptional regulation of AP2-G and downstream target genes[55]. The mGRN analysis enabled us to predict that AP2-G5 interacts with additional target genes, including *PMIII* (PF3D7_1408100), *ThrRS* (PF3D7_1126000), and *MRP2* (PF3D7_1229100), these genes were associated with GO terms implicated in response to drug or xenobiotic stimuli (Supplementary Data 5), as well as genes related to hemoglobin catabolic processes, such as *PM4* (PF3D7_1407800), *UBA1* (PF3D7_1333200), and a canonical male gametocyte marker, the dynein-heavy chain protein-encoding gene (PF3D7_1122900) (Supplementary Data 5). In agreement with ChIP-seq analysis from a previous study[55], AP2-G5 binds to DNA-binding sites upstream of the AP2-O3 (PF3D7_1429200) and NUP116 (PF3D7_1473700) TSSs. AP2-O3 regulates essential genes in the female gametocyte maturation from stage II to III and is implicated in the elongation of the invasive stages in the mosquito midgut[56]. In contrast, NUP116 is associated with GO terms implicated in protein import to the nucleus and as a co-regulator of transcription. Further, we observed that PIC5 (PF3D7_1310700), a PhIL1 interacting candidate of IMC encoding genes, is predicted to be repressed by AP2-G5 (Supplementary Fig. 9B; Supplementary Data 5). The unannotated ApiAP2 gene, PF3D7_0420300, was also found in Mod A and is predicted to regulate

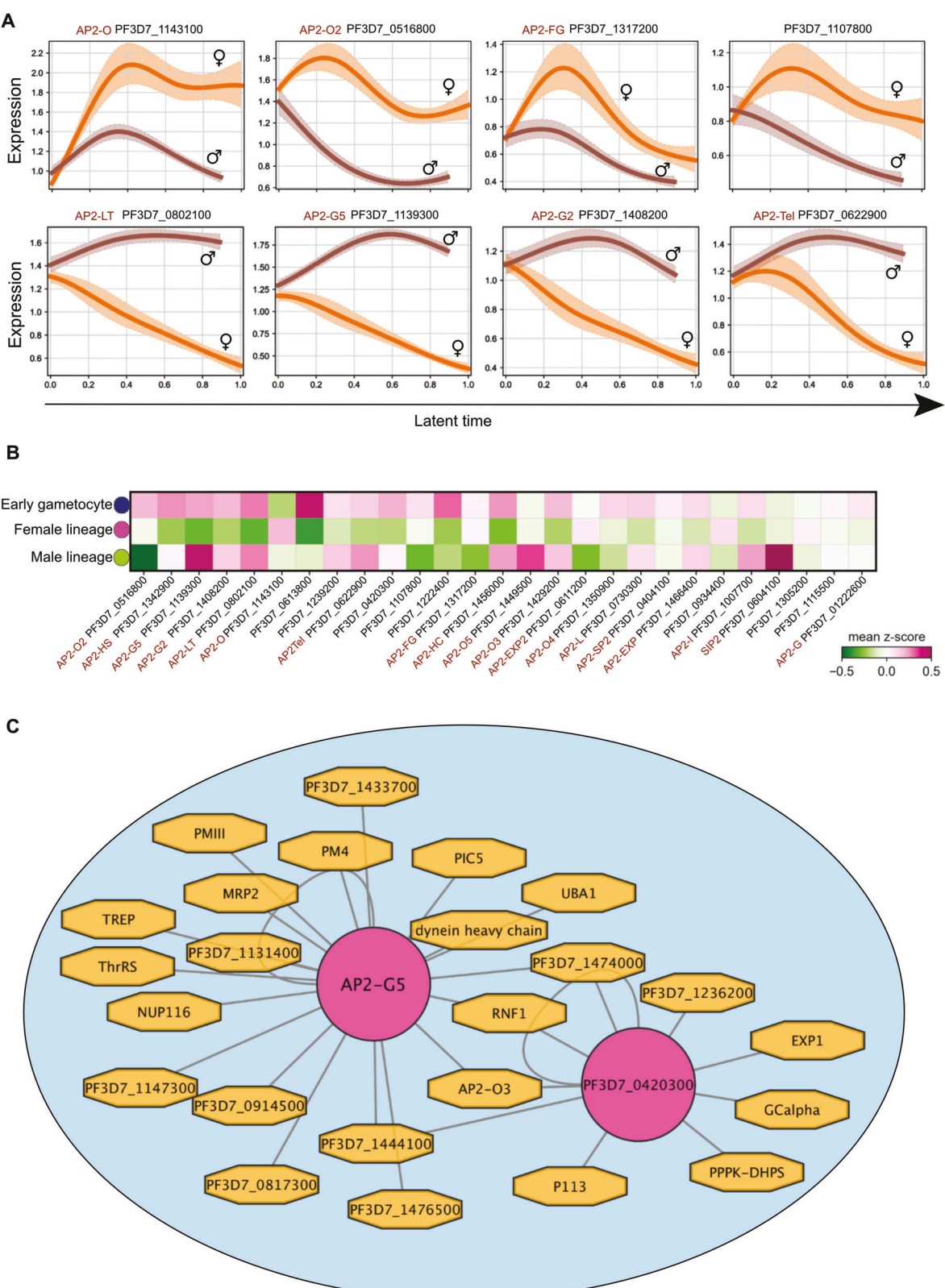

**Fig. 5 | Motif-driven gene-regulatory network analysis of *P. falciparum* gametocytogenesis. A** Modeled gene expression trends of selected ApiAP2 transcription factors toward the terminal state of mature female (top panel) or mature male gametocytes (bottom panel) plotted against the latent time (X-axis) and expression level (Y-axis). The shading represents the fate probabilities of gene expression with a 95% confidence interval and margins of 0.015. **B** Heatmap depicts estimation of mean Z-score expression of ApiAP2 transcription factors across the annotated cell classes (states) generated using DDRTree classification. **C** The Motif-driven Gene

Regulatory Network (mGRN) in module A (Mod A) was generated as an output of the Area Under the Curve (AUC) motif enrichment in single cells, inferring the weight-based interactions between the TF and the target genes. The analyses were performed using the pySCENIC default parameters. Pink circles depict nodes of TFs and their predicted target genes in marigold. The distance of the edges depicts the estimated weight of co-expression. The predicted target genes for each regulator were manually validated using PlasmoDB, where DNA motif patterns 1.5 kb upstream of the TSS region were analyzed.

an array of potential target genes, among them, surface protein P113 (PF3D7_1420700), and EXP1 (PF3D7_1121600), both are parasitophorous vacuole proteins important for the protein export machinery and nutrient uptake (Fig. 5C; Supplementary Fig. 9A, B)[57,58]. This could be an indicator of its involvement in regulating genes required for protein export and channel permeability during gametocyte development. In addition, we observed the unannotated ApiAP2 PF3D7_1222400 and PF3D7_1107800 in Mod B, with microtubule structure component alpha-tubulin 1 (PF3D7_0903700) potentially regulated by both transcription factors, likely to maintain the gametocyte cytoskeletal structure (Supplementary Fig. 9C)[59]. In Mod C we found the unannotated ApiAP2, PF3D7_1239200, with relatively elevated expression in the early gametocyte and the male lineage (Fig. 5B; Supplementary Fig. 9C), interacting with two unannotated genes. We observed AP2-G2 (PF3D7_1408200) and AP2-O4 (PF3D7_1350900) in Mod E and Mod D, respectively (Supplementary Fig. 9C). In line with previous studies[16], AP2-G2 is predicted as a repressor in our regulon analyses (Supplementary Fig. 9B) and appears to repress the expression of its putative target genes including *AHA1* (PF3D7_0306200) and PF3D7_0209500 in the male lineage. These two genes are indicated to be downregulated in their expression in C5. AHA1 is implicated in ATPase activation of HSP90 and protein folding based on the GO terms and is required for maintaining homeostasis during gametocytogenesis[60]. Finally, we predict AP2-O4 to act as a repressor of *SIP2* (PF3D7_0604100) in Mod D (Supplementary Fig. 9A–C). AP2-O4, which has been described to be implicated in ookinete development in the mosquito, does not indicate a sex-specific correlation (Supplementary Fig. 8). However, we found its "repressing" regulon enriched in the male lineage (Supplementary Fig. 9B, C).

## Discussion

Gametocytes are notoriously difficult to study due to their relatively low abundance in the blood and their variable and asynchronous development, and traditional bulk RNA sequencing approaches cannot capture the heterogeneity of differential male and female gametocyte development. In recent years, single-cell transcriptomics has emerged as a powerful tool to overcome these challenges and enable characterization of gametocyte development at high resolution[18,19]. In this study, the fluorescent *P. falciparum* NF54 Peg4 tdTomato line was used for stage-specific enrichment of gametocytes through FACS, followed by scRNA-seq and computational analysis to investigate the gene expression programs underlying male and female lineage development in *P. falciparum* gametocytes. We used previously reported canonical markers of mature gametocytes to confirm the mapping of differentiated mature male and female gametocyte cell populations in our single-cell dataset.

Through gene expression correlation and splicing kinetic analyses, we identified a set of putative driver genes that exhibit continuous expression trends towards either male or female mature gametocyte cell populations. In addition, we experimentally validated the presence of immature mRNA of the selected putative driver genes between early and late gametocyte clusters. Through ranking the correlation of continuous expression towards mature male or female clusters, we filtered lineage-specific genes based on 3'-end exon-intron sequence coverage and the presence of immature transcripts among the early stages of the developing gametocytes. Our final list yielded a total of 84 putative driver genes of the female lineage and 56 putative driver genes of the male lineage. Subsequently, we sought to further identify the putative driver genes using PlasmoDB, which indicated that the majority of genes have no functional annotation, suggesting that many of the processes during gametocytogenesis are not yet revealed.

Thus, we performed a series of analyses to cross-validate the putative driver genes firstly by performing rt-qPCR on bulk RNA samples from early, mid and late timepoints of development. We utilized the smart-seq2 chemistry on individually captured gametocytes followed by rt-qPCR. Our results reinforce that the majority of our top-ranked male and female driver genes display sex-specific expression. The male and female expression patterns in individually isolated gametocytes highlight the potential of these putative driver genes in serving as early-stage sexual determinants. In particular, MD1, MD3, and MD8 in males and FD2-3 and FD7-8 in females stand out as particularly interesting marker genes for identification of early male or female gametocytes due to their apparent reproducibility and specificity both computationally and experimentally. MD7 is annotated as a putative MORN repeat protein in PlasmoDB, it is expressed in all males but also in 27% of the female gametocytes, which may indicate a more general role linked to gametocyte development. FD4-6 are expressed in a large number of female gametocytes (73–92%) but also appear to be expressed in some of the male gametocytes. This could indicate a more general role for these three genes in gametocyte development or that there is a common expression in some male and female gametocytes representing a transient early state of development. Future studies are necessary to specify the function of these lineage-specific genes as well as evaluate their potential as novel drug targets to inhibit malaria transmission.

Next, these genes were assessed in terms of their fitness from a broader evolutionary perspective given that genes that are crucial for survival and with limited redundancy are prime targets for disease control strategies. Sex-driving genes are frequently under positive selection[61] and targeting these genes tend to have reduced off-target effects given the species-specific function of many sex genes. The putative male and female driver genes identified here were potentially under more frequent positive selection in comparison to control genes. These findings indicate that sex genes could be promising targets for intervention strategies given the essential nature of sex genes and species propagation on the evolutionary stability of the organism[61–63].

Male gametes have an elevated requirement of energy production due to their rapid DNA replication followed by exflagellation. The RNA velocity analysis of the male lineage revealed an elevated level of unspliced mRNA in the lactate dehydrogenase (LDH) gene, PF3D7_1325200. LDH has been shown to have high expression during the asexual erythrocyte and ookinete stages, with increased mRNA levels in gametocytes and which peak during the ookinete stage[45,64]. LDHs play important roles in the regulation of glycolysis and oxidation/reduction in malaria parasites[65,66]. The skewed expression of this gene in male gametocytes is likely linked to the elevated energy requirements in the highly motile microgamete. In addition, our findings showed that PfSIP2 is enriched in the male lineage. We speculate that PfSIP2 is involved in heterochromatin formation during the rapid DNA replication that occurs in micro-gametogenesis, highlighting the necessity of further functional investigation of these two proteins.

The HDP1 transcription factor has been shown to promote the expression of IMC genes and PhIL1 which are necessary for gametocyte morphological changes[37]. We went on to specifically investigate IMC genes such as IMC1c, IMC1d, and IMC1h and the IMC-associated gene PhIL1. Our pseudotime analyses indicate that HDP1 precedes the expression of the IMC genes and that these genes are over-represented in the female gametocyte lineage compared to the male, suggesting a differential machinery in achieving developmental morphological modification in female *vs* male gametocytes. Interestingly, PIC5, a PhIL1-interacting candidate was found to be downregulated in the mature male gametocyte, suggesting a potential mechanism of inhibition of a subset of IMC-associated genes in the male lineage.

Notably, deconvolution analysis of our single-cell dataset and the bulk gametocyte transcriptome time course, established by van Biljon and colleagues[34], suggests that the expression of female lineage transcriptome occurs as early as day 0 of sexually committed rings and

trophozoites within the bulk samples. This was previously overlooked in single-cell studies due to limitations in sample size and lack of computational tools to accurately infer sexual cell fate probabilities and trajectories between initial and terminal states. In this study, we identified an early signal of the male lineage transcriptome occurring as early as day 4 (stage I to II) in the bulk transcriptome dataset. However, further research is necessary to characterize the early gametocyte clusters associated with male and female development.

During sexual commitment, an array of AP2-G target genes, including a subset of ApiAP2 transcription factors are activated within hours of early gametocyte development[67]. We investigated lineage-specific expression trends of members of the ApiAP2 family. AP2-O, AP2-O2, and AP2-FG were previously shown to be expressed in mature female gametocytes[33,68,69]. Our lineage tracing analysis supports these findings and provides an additional AP2 gene namely, PF3D7_1107800, which has not been linked to female gametocyte development before.

ChIP-seq experiments indicate that upstream binding of HDP1 includes IMC genes and an additional two ApiAP2 transcription factors that are expressed during the late gametocyte stages[37]. We identified AP2-O2 (PF3D7_0516800) that is correlated to the female gametocyte lineage with its expression peaking within the late stages of developing female gametocytes, and it is required for the regulation of transcripts essential for ookinete development in *P. berghei*[33]. In *P. falciparum*, AP2-O2 is shown to be one of the HDP1-regulated genes together with the components of inner-membrane complex IMC and PhIL1 that are temporally expressed during mid-to-late stages of the gametocytogenesis[37]. *Pb*AP2-FG has been shown to activate the transcriptional divergence of the female gametocyte lineage in *P. berghei*[70]. It has further been shown that disruption of *Pb*AP2-FG in *P. berghei* results in deformed female gametocytes and is suggested to activate most of the female-specific genes[17].

AP2-G2, AP2-G5, and AP2-Tel (AP2-SP3) were found to be specifically associated with the male gametocyte lineage. Previous studies have identified that the AP2-G2 promoter is bound by AP2-G, which represses the expression of asexual proliferation genes. AP2-G2 and AP2-G5 have shown to be integral for the regulation of gametocyte production in *P. falciparum*[16,55,71]. We anticipate that AP2-G5 represses PIC5 in the male lineage, hence the finding that *PIC5* and a set of IMC genes are downregulated in C5 and the male lineage in comparison to the female (Figs. 1C and 2G). Previous reports indicate that aberration or deletion of AP2-G2 substantially hampers male gametocyte production with vast downregulation of male development gene 1 (MDV1, PF3D7_1216500), an essential gene in the production of male gametocytes[72]. On the other hand, AP2-G2 holds a conserved evolutionary role in *Plasmodium spp.* gametocyte development and is suggested to act as a bi-functional regulator, either by activating late gametocyte genes beyond stage III or forming a complex with H3K36me2&3 to establish a coordinated cascade of repression between stage II-III gametocytes[21]. Here, we show that AP2-G2 is correlated with the male gametocyte lineage and that its expression profile suggests that it regulates at least two genes, of which one (PF3D7_0209500) has not yet been functionally annotated. The second is an annotated activator of heat shock protein 90, ATPase (AHA1) that is possibly exerting a stimulatory effect on the ATPase activity of Hsp90 and protein folding during male gametocyte development[71].

PF3D7_0802100 (AP2-LT) has previously not been described to be involved during gametocyte development, it is a component of the SAGA-like activator complex, which was previously shown to be implicated in erythrocyte invasion and virulence during the asexual stages[73]. Our data indicates that AP2-LT is expressed in the male lineage and we found that its binding motif, ACACAC, is located within a 1.5 kb region upstream of the TSS of; ADF2, MA, CDPK1, PMIX. CDPK1 has previously been shown to play a significant role during gametogenesis in the mosquito, where mature males from a CDPK1 KO line were unable to exflagellate[74].

An additional member of the AP2 family predicted from our data to be associated with the male lineage, AP2-Tel/SP3, is a component of the *P. falciparum* telomere-binding protein complex implicated in telomere biology through binding to all 14 chromosomes[75]. AP2-Tel/SP3 is expressed throughout the IDC while its ortholog in *P. berghei* (AP2-SP3) has been shown to be associated with sporozoite release from the oocyst[33]. Recent findings in *P. falciparum* single-cell data showed the expression of egress-related genes preceded by AP2-Tel/SP3 in sporozoites, suggesting a global activation of egress genes that are potentially required in gametogenesis and sporogony[76].

These new findings provide essential information about members of the ApiAP2-family and reveal their lineage-specific associations linked to the transcriptional programs of male or female gametocyte development.

Mapping the gene regulatory network (GRN) during gametocytogenesis is a useful tool to identify potential upstream regulators and their acting *cis*-regulatory elements implicated in male or female lineage development. Nevertheless, GRN comes with limitations as the interacting genes (nodes) are based on co-expression patterns and do not take into account the causative factor of the interaction[77]. Here we address this limitation by implementing a motif-driven component into the gene regulatory network analysis through pruning these interactions based on the detection of DNA genomic binding sites in the upstream regions of the target genes. The upstream regulator is linked to its downstream target gene via its binding site in the *cis*-regulatory region. We custom-built a *cis*-target motif dataset for *P. falciparum* using position weight matrices from Campbell et al.[25]. We then used the mGRN dataset to infer regulons, i.e. groups of genes regulated by a common regulatory protein, enriched in our single-cell dataset. We demonstrate gene modules (Mod A through Mod E) that are composed of ApiAP2 gene family members, where some have not previously been linked to particular stages in the *P. falciparum* life cycle. Previous data from Shang and colleagues[55] showed the downregulation of AP2-G and a set of early gametocyte genes in *P. falciparum* AP2-G5 knockout parasites. Our findings identify a subset of the AP2-G5 target genes (AP2-O3 and NUP116), with additional genes including PMIII, PM4, and dynein-heavy chain, predicted to be regulated by AP2-G5 during gametocytogenesis. The potential regulation of the male gametocyte canonical marker dynein-heavy chain (PF3D7_1122900) by AP2-G5 suggests a lineage-specific downstream regulation of target genes. Moreover, *PIC5* was observed among potential AP2-G5 target genes, suggesting the regulation of inner membrane pellicle complex-associated IMC genes. Our differential gene expression analyses indicated that *PIC5* is downregulated in the male gametocyte cluster (C5), while the regulon analysis of AP2-G5 points to a repressive effect on *PIC5*. Therefore, we hypothesize a potential repressing effect on *PIC5* by AP2-G5 particularly in the male gametocyte lineage, suggesting a lineage-specific gene regulation controlled by AP2-G5. In the same module (Mod A), the non-annotated PF3D7_0420300 gene was observed, suggesting that it may be a co-regulator with AP2-G5, since we found shared putative target genes regulated by both transcription factors including; AP2-O3, RNF1 (PF3D7_0314700), and two additional non-annotated genes (PF3D7_1444100, PF3D7_1474000). Further, the AP2 gene, PF3D7_0420300, interacts with the potential target gene *P113*, which is known to encode a membrane-anchored protein. Interestingly, this gene was recently suggested by Campeotto and colleagues as a vaccine target[78,79].

In summary, the most common form of sex determination in animals involves males and females, eukaryotic microbes however, are known to display a variety of mating types, commonly at defined points in their complex life cycles. Eukaryotic microbes provide unique opportunities to study the evolution of sexual recombination among eukaryotes in general[80] and may serve as a foundation to study the evolution of sex, including why and how it evolved in the first place[80].

Members of the apicomplexan phylum generally perform sexual recombination as part of their life cycles, where mating types include male and female gametes but where sex is not determined at the chromosomal level.

In the context of sex allocation and gametocyte density, natural populations of *P. falciparum* have been reported to have an increase in sex ratio when gametocyte densities decrease[81,82]. Our findings may bring further context to the transcriptional regulation of sex ratio allocation which is either held in single genotype or dry season conditions, or which are not held depending on mixed genotype, wet season conditions or treatment with antimalarials[81,82]. The highly regulated sex-dependent transcriptional programs explored in this study that lead to male and female differentiation in *P. falciparum* may offer evolutionary context to the functional development of tightly packaged sex-related chromosomes, given that epigenetics has been accredited with the evolution of sex chromosomes[83] and our findings, as well as others, highlight the involvement of several genes known to be regulated epigenetically[21,22]. Perhaps separate sex chromosome function evolved through a necessity for coordinating sexual differentiation in multicellular organisms, as opposed to that of unicellular organisms.

In total, our study provides a deeper understanding of the transcriptional programs and gene regulatory pathways involved during male and female *P. falciparum* gametocyte development. The putative male and female lineage driver genes remain to be functionally characterized and further research is necessary to fully comprehend their role in contributing to sexual development and cell fate. In order to achieve a more thorough comprehension of gametocyte biology, it is imperative to further validate the regulatory roles of the AP2 family members in cell fate determination. In addition, investigating factors known to have a key role in *P. falciparum* differentiation and sexual-stage development, such as chromatin accessibility and histone modifications is an evident avenue for future research and is necessary to further deepen our understanding of regulatory elements underlying parasite sexual cell fate. Such insights will have far-reaching implications for comprehending the evolution and adaptation of regulatory factors linked to the differentiation of mating types and sex, among a large array of eukaryotic organisms. In addition, our study together with the recent studies by Gomes et al. and Russel et al.[18,19] provide the malaria community with essential novel genetic markers for malaria transmission blocking strategies.

## Methods

### In vitro culture of *P. falciparum* parasites and induction of gametocytes

The gametocyte-producing NF54 Peg4-tdTomato transgenic cell line[24] was used in this study. Parasites were cultured in T75 tissue culture flask at optimal growth conditions, including RPMI supplemented with 10% heat-inactivated type A human serum (Karolinska Hospital blood bank, Stockholm, Sweden) and 5% hematocrit, using blood from type O positive donors[84]. Cultures were individually gassed with 96% $N_2$, 1% $O_2$, and 3% $CO_2$ and maintained at 37 °C on a shaker (50 rpm). Parasites were maintained at ≤1% parasitemia and were monitored daily before the initiation of gametocyte induction. Gametocyte inductions were performed at 10–12% parasitemia, using the commitment assay described in Fivelman et al.[85]. Briefly, synchronized parasites were cultured to high parasitemia of rings (day −2). The media was partially replenished on day −1 as the parasites reached the trophozoite stage, finally the parasite cultures were split 1:5 prior to the end of the 48 h cycle. From day 0 (gametocyte rings), the parasites were cultured in the presence of 50 mM (final concentration) N-Acetyl-D-glucosamine (Sigma-Aldrich, cat. no. A3286) in order to remove asexually replicating parasites from the culture[85]. Cultures were synchronized and induced at 48 h intervals, to enable sorting of all timepoints on the same day. The cells were maintained at optimal gas and culture conditions by placing capped T75 culture flasks in a Cellbox shipper (Cellbox, Hamburg, Germany) until enrichment by FACS. Production of gametocytes for FACS enrichment and downstream molecular experiments was repeated once for a total of two replicates.

### Gating strategy for gametocyte enrichment using FACS-based cell sorting

Using the *P. falciparum* NF54-tdTomato line, we included four consecutive timepoints (Days 1, 3, 5, and 7) in order to cover gametocyte maturation from the very early stages of development. The culture samples were then purified using Magnetic Activated Cell Sorting (MACS) on a MidiMACS™ (Miltenyi Biotech, Switzerland), for the day 3 to day 7 gametocytes, in order to remove asexual stage parasites and non-infected RBCs. On the day of sorting, gametocytes were counted using a hemocytometer and the gametocytemia was set to 400,000 cells/ml and maintained at optimal growth conditions, gassed, and placed in a CellBox portable incubator until the time of sorting. Prior to cell sorting, 2 µg/ml of Hoechst live cell stain 33342 was added to the gametocyte cultures and incubated for 15 min. The sorting was performed with a MoFlo Astrios EQ (Beckman Coulter, USA) cell sorter using 355 and 532 nm lasers for excitation, 100 µm nozzle, sheath pressure of 25 psi and 0.1 µm sterile filtered 1xPBS as sheath fluid. Flow sorting data was interpreted and displayed using the associated software Summit v 6.3.1. Hoechst was used to differentiate between asexuals and gametocytes and cells were sorted into a 1.5 ml Eppendorf tube using the purify mode 1–2 drop envelope. The sample and receptacle holder were cooled at 4 °C during the sort. Side scatter was used as trigger and cells containing gametocytes expressing tdTomato were gated on increasing red fluorescence in a bivariate plot vs DNA staining of Hoechst (532-571/21 vs 355-448/59) in combination with a "cells gate" using scatter (SSC vs FCS), and 400 events were set for each time point. We used a gating strategy with the ambition to include gametocyte-fated rings in the early time point (Day 1), based on the detection of Hoechst using the 355 nm laser and a relatively lower level of tdTomato fluorescence detected using the 532 nm laser (Supplementary Fig. 1A). We sorted -10,000 cells from each time point into individual tubes. The final four tubes of gametocytes were then pooled into a single tube for a final concentration of 20,000 cells, followed by sample loading onto a 10X Chromium Next GEM Chip. To ensure the reproducibility of the gametocyte enrichment strategy, this method was performed twice in biological duplicates using the predefined FACS settings mentioned above for all the data points collected (Fig. 1A).

### Library preparation and sequencing

10X Chromium single-cell 3' reagent kits (V3) (10X Genomics) were used according to the manufacturer's instructions (CG000183 Rev A), aiming for 4000–8000 cells per library. Library quality control and quantification were performed using a KAPA Library Quantification kit for Illumina platforms (Kapa Biosystems, Cape Town; South Africa) and a 2100 Bioanalyzer High Sensitivity DNA kit (Agilent, Santa Clara, USA). Libraries were sequenced in an Illumina HiSeq 2500 on Rapid Run settings using asymmetric paired-end sequencing (26 cycles for Read 1 and 98 cycles for Read 2) and pair-ended 150 bp (PE 150) reads that yielded approximately 160 million reads per run.

### Genome mapping and generation of expression matrices for scRNA-seq data

Cellranger (version 7.1.0) was used to create a reference transcriptome file from the reference genome of *Plasmodium falciparum* version 50 (ASM276v2) using standard parameters of the "mkref" function implemented in the pipeline. The raw sequencing fastq files were demultiplexed and then passed to the Cellranger pipeline using standard parameters of the function "count" to assign reads to the cell barcode. The output files containing raw processed gene expression

matrices and barcoded information were used for downstream analysis. Preliminary quality control metrics were yielded spontaneously from the cellranger pipeline analysis giving a total of 7320 cells. Initial computational QC with median genes per cell is 986 with mean reads per cell 258,050.

## Data processing for dimensionality reduction

Further quality control and downstream measurements were processed using the ScanPy pipeline (V 1.9.1). We used the Cellranger generated "raw_feature_bc_matrix" as the input file for the Scanpy pipeline to filter out individual cells containing fewer than 200 genes and more than 3000 genes in a minimum of two cells. Finally, 4555 cells were obtained for downstream processing. The resulting count matrix was a library size normalized to $10^4$ transcripts; the data were log-transformed with a pseudocount equal to 1. Next, highly variable genes with mean log expression between 0.0125 and 3 with minimum dispersion of 0.5 were selected using "scanpy.pp.highly_variable_genes" for further downstream processing. The number of read counts was then regressed using "scanpy.pp.regress.out", the data was then scaled with "scanpy.pp.scale". Batch correction was attempted using BBKNN[86] but no noticeable improvement was seen (Supplementary Fig. 1F). We thus chose to use the raw data in order to avoid adding bias. Clustering (described in the next section) is also not driven by batch effects (Supplementary Fig. 1G).

## Visualization of cluster-wise scRNA-seq using UMAP

We first performed a principal component analysis (PCA) and selected the number of top principal components (PCs) with the highest explained variances based on the "elbow plot" of the screen plot of the PCs. In practice, we found that the first 20 principal components are quite robust to construct a directed graph connecting the nearest neighbor cells in the reduced PCA space, using the "scanpy.tl.umap" which is a convenient wrapper of the original function 'UMAP' from the Python package 'umap-learn'. The directed graph was then converted to undirected, to obtain 2D and 3D UMAPs. UMAP is a graph-based manifold learning method that can provide good visualization while preserving the intrinsic structure of the original data. Meanwhile, it is also a computationally efficient tool for large-scale datasets. It first computes the approximate k nearest neighbors (kNNs) for each data point, building a weighted mutual-kNN graph with each node representing each data point (cells in our case), and then embeds each node of the graph into the lower dimensional space.

## Clustering of cell populations and identification of differentially expressed genes

To cluster single cells into distinct populations, we used a graph-based clustering approach, which applies Louvain[87] community detection on the weighted kNN graph built by UMAP. The Louvain algorithm is very similar to the Leiden community detection algorithm which is widely used for single-cell clustering[88]. This clustering method was achieved by the built-in interface 'scanpy.tl.louvain' from ScanPy, with the parameter 'resolution' equal to 1. To get a solid clustering result, we compared the DEGs of each cluster. The DEGs were found using the Wilcoxon rank test and adjusted with Bonferroni correction, which computes highly differentially expressed genes in each cluster. This was achieved by the function 'scanpy.tl.rank_genes_groups' in the ScanPy library.

## Discriminative dimensionality reduction (DDRTree) algorithm for lineage annotation

We adopted the decision tree-based classification implemented in Monocle 2[89] (V 2.4.0) since it allows us to capture the bifurcation of cell fate decisions of putative lineage branch points during the process of ordering the cells along the developmental trajectory. To construct the single-cell trajectory, we imported a normalized, scaled count matrix

generated by ScanPy analysis and used Louvain clusters to order cells in a lower dimensional space. We then applied a reverse graph-based embedding algorithm to learn the structure of the trajectory with unsupervised analysis. Significant genes with $q < 0.01$ were selected from all identified states with a total number of 4555 single cells. Discriminative dimensionality reduction (DDRTree) was performed with regression in the UMI counts to eliminate unwanted variation introduced by the depth of sequencing between samples, and cells were ordered along the trajectory according to their pseudotime value. The number of genes and UMIs for each of the clusters were tested and passed our predefined cut off of 200 genes expressed per single cell to be included in the study. The low UMI was used in order to allow the inclusion of gametocyte rings, which are known to show low global gene expression.

## RNA velocity and cell fate probability prediction

To interpret the global transcriptional progression of gametocytes and their cell fate decision, we established the cell continuum of cell differentiation, data layers of unspliced and spliced mRNA for the entire data generated with Velocyto CLI (v.0.17.17) according to the CLI usage guide (Velocyto run10x). The output loom files were combined using "loompy". The merged loom file was imported into the scVelo package (v1.0.6)[90,91]. The unspliced and spliced mRNA counts of cells from clusters C0-C9 were extracted. We used the "merged.utilis" function in the scVelo pipeline, where cells with low pre-mRNA counts were filtered out. In short, the gene-specific velocities are obtained by fitting a ratio between unspliced and spliced mRNA abundances and then computing how the observed abundances change from those observed in a steady state. The ratio of 'spliced', 'unspliced', and 'ambiguous' transcripts were calculated and data were pre-processed using functions for detection of minimum count number, filtering and normalization using "scv.pp.filter_and_normalise" and followed by "scv.pp.moments" function. The gene-specific velocities were then calculated using "scv.tl.velocity" with mode set to "deterministic" and "scv.tl.velocity_graph" function to generate a velocity graph, and visualized using the "scv.pl.velocity_graph" function. In addition, we used the "scv.tl.recover_latent_time" function to infer a shared latent time from splicing dynamics and plotted the genes along the time axis sorted by expression along dynamics using the "scv.pl.heatmap" function.

To identify putative gene programs that potentially drive male and female gametocyte progression, a directed transition matrix was computed, coarse-grained into microstates to infer latent-space initial and terminal states. We then selected known male and female marker genes to model the expression towards either inferred terminal state using the generalized additive model "GAM" (see Code availability section). Importantly, no previously known driver genes have been shown to be involved in gametocyte sex-specific lineage progression. By correlating gene expression trends with the enrichment of an unspliced mRNA reads ratio, we sorted lineage-specific gene programs according to correlation value (Supplementary Data 4).

For partition-based graph abstraction (PAGA) analysis[92], we used the standard parameters in ScanPy implementation, "sc.tl.draw_graph" (init_pos = 'paga') followed by "sc.tl.paga" (threshold = 0.3) and "sc.pl.paga" function for analysis and plotting, and similarly, scVelo implementation function "scv.tl.paga" with default parameters and "scv.pl.paga function" for velocity-driven PAGA analysis and plotting.

## Data integration with previously published transcriptome datasets

We integrated our scRNA-seq data with the bulk RNA-seq data from the van Biljon et al. study[34] to chart the correlation of the real-time gametocyte stages transcriptional change. In this analysis, we followed a deconvolution linear regression model to calculate the cell type proportion using the shared expression profile between the

bulk RNA-seq data and our single-cell data. We then fit the regression model to transfer the cell type labels (stage and time point identifiers from the van Biljon bulk dataset) into our single-cell data and aligned the transferred labels on our pre-estimated pseudotime trajectory. Note that our bulk deconvolution analysis initially included all the bulk transcriptome data points from the van Biljon study, however, we were not able to achieve optimal "label transfer" due to the variability of the transcriptome in the early and late collected stages in the bulk data. This was similarly described by Gomes and colleagues[18] in their study calculating the expression correlation between the bulk and single-cell data. Upon several iterations of the deconvolution analysis and considering the noise introduced by the bulk RNA-seq data, we imputed the bulk data to remove outliers from each time point or stage. We subset the bulk data from day 3 to day 7 and further projected the transferred labels across our single-cell data PCA coordinates (Supplementary Fig. 2C).

In addition, we performed integration analysis with the Gomes et al. dataset, firstly including their single-cell data from the WT parasite line. We used the Scanpy external function for gene expression matrix correction (Scanorama v1.7.4)[26], which relies on correction of the sequencing depth to obtain scalable gene expression profiles across the single cells from different datasets. We calculated the top 2000 highly variable genes for each batch and unified the variable genes to subsequently calculate the dimensionality reduction for the combined and integrated datasets.

Second, we integrated our dataset with the Gomes et al. single-cell dataset from their md1 knockout parasite line, using the same computational method to further validate the male gametocyte population in our dataset. Subsequently, we calculated the percentages of early, male and female cells in both datasets (Supplementary Fig. 3).

## Transcription factors and regulon activity inference using pySCENIC

The analysis of regulon activity was performed following the standard pySCENIC protocol[53] (v 012.1). Binned (bin 50) or segmented cell expression data matrices were used as the input to pySCENIC. The pipeline has three steps: (1) identification of potential TF targets based on co-expression; (2) TF-motif enrichment analysis to identify the direct targets (regulons); and (3) scoring the activity of regulons (or other gene sets) on single cell types. The expression matrix was subjected to GRNBoost2 algorithm analysis to reconstruct the coexpressed gene network for each ApiAP2 transcription factor. The transcription factor co-expression modules were then analyzed by RcisTarget and their potential targets were further filtered with default parameters. The filtered potential targets were used to build the regulons. The regulon activity AUC (Area Under the Curve) was analyzed with AUCell and the active regulons were determined with the AUCell default threshold. The activity of regulons for each bin was then mapped to the DDRTree-based cell class (Early gametocyte, female lineage, and male lineage). The gene network for transcription factors and identified target genes were sorted according to the weight of interaction. In brief, the raw processed expression matrix for all genes and all regulon activity matrices were used as input. For the gene modules, the data were normalized by the total UMI number of each bin, from which the kNN graph of genes was created using the *create_knn_graph* function with the parameters: *n_neighbors = 30* (for regulon, *n_neighbors = 10*), then genes or regulons with significant autocorrelation (*FDR < 0.05*) were kept for further analysis. The modules were identified using the *create_modules* with the parameters: *min_gene_threshold = 20* and *fdr_threshold = 0.05* (for the regulon: *min_gene_threshold = 5* and *fdr_threshold = 0.05*). mGRN was then visualized and annotated using Cytoscape[54].

## RNA Isoform prediction analysis

The transcript prediction was performed separately for each identified cluster using Stringtie2[93] (v 2.2.1). Mapping files were XS-tagged and divided by clusters. Each of the files was run separately using Stringtie2 with default parameters. Transcript predictions were then compared and collapsed across samples into a Stringtie GTF for each cluster using custom scripts. Finally, we compared the predicted transcripts sequences to those annotated in the *P. falciparum* genome annotation file (v50). Custom tracks were then visualized with the IGV web app (v 1.12.8). All predicted isoforms are available for interactive browsing using IGV web at https://igv.org/app/ and the files are available at Zenodo, https://zenodo.org/deposit/7652581.

## Validation of RNA isoforms using rt-qPCR

A qPCR experiment was designed to show the presence of mature and immature forms of mRNA during the different *P. falciparum* gametocyte developmental stages in order to validate the RNA velocity outcome. Total RNA was extracted from Schizonts and early and late-stage gametocytes from the *P. falciparum* NF54-peg4-tdTomato line. The cDNA was synthesized with the Transcriptor First Strand cDNA Synthesis Kit (Roche, Basileia CH) using the oligo dT primers and following the manufacturer's instructions. Female putative driver genes including; PF3D7_1338800, PF3D7_1466500, PF3D7_0418800, PF3D7_0924800, and PF3D7_1331600, were selected, based on the coverage of their last intron on the 3'-end of the ORF. The 18S ribosomal RNA was used as a housekeeping gene. Three (two for 18S) primers (Supplementary Table 1) were designed for each target gene using the Geneious prime software (Dotmatics, Boston, USA). The reverse primer was always designed to anneal to the last exon on the 3'-end of each gene whereas two forward primers were independently designed to anneal to the exonic (EFW) or intronic (IFW) regions. The cDNA samples were diluted to 20 ng/µl and all real-time PCR reactions were performed using the QuantStudio5 (Thermo Fisher Scientific, Waltham USA) using the Power Up SYBR Green Master Mix (Thermo Fisher Scientific, Waltham). The experiments were carried out in duplicate for each data point. The relative quantification in gene expression was determined using the $2^{-\Delta\Delta Ct}$ method[94]. The data were normalized to the internal control 18S and Schizont cDNA was used as a calibrator.

## RNA extraction and cDNA synthesis for rt-qPCR assays

Asexually replicating parasites (rings, trophozoites, and schizonts) and synchronous gametocyte cultures on days 2, 5, and 9, from the NF54 Peg4-tdTomato line were produced. Parasites were harvested and sedimented at 1500 g for 5 min at 4 °C, followed by a single wash in sterile 1x DPBS. The cells were then resuspended and diluted at a ratio of 1:10,000 in RNA later stabilization solution (Thermo Scientific, Waltham, Massachusetts, USA).

The Agilent High Sensitivity DNA Kit (Santa Clara, California, USA) was employed to quantify cDNA concentrations on an Agilent 2100 Bioanalyzer (Santa Clara, California, USA). Subsequently, all samples were diluted to a final concentration of 15 pM/µL. Total RNA was extracted using TRI-reagent (Sigma-Aldrich, cat. no. AM9738). In short, RNA samples were treated with DNAse (New England Biolabs, cat. no. M0303S) for 30 min at 37 °C. cDNA was synthesized using the Transcriptor First Strand cDNA Synthesis Kit (Roche, cat. no. 04379012001).

## Gametocyte isolation by capillary-based micromanipulation

Individual gametocytes were isolated by micromanipulation on days 2 and 8 of development, using the Eppendorf TransferMan NK2 in a Leica DMi8 inverted wide-field fluorescence microscope (Leica, Germany), allowing the detection of tdTomato-expressing cells. Collected cells were transferred to 0.5 µL 1x DPBS and subsequently transferred to a 0.2 mL thin-walled microcentrifuge tube containing 2.3 µL lysis

buffer, composed of 0.8% Triton X-100 (Sigma-Aldrich, cat. no. T9284), 1:20 RNAse Inhibitor (Clontech, Cat. no. 2313 A), 1 μL 10 mM dNTP mix (Fermentas, cat. no. R0192), and 1 μL of the *Plasmodium*-optimized Oligo-dT30VT primer (5'-AAGCAGTGGTATCAACGCA-GAGTACT30VT-3') diluted to 10 μM[95]. Tubes were snap-frozen using liquid nitrogen and stored at −80 °C until further use.

## Single-cell cDNA synthesis and amplification using smart-seq2
cDNA synthesis and amplification from individual parasites was performed using a modified version of the Smart-seq2 protocol[4]. In summary, cDNA synthesis included the Oligo-dT30VT primer, followed by amplification over 24 cycles. Amplified cDNA products underwent purification using Agencourt Ampure XP beads (Beckman Coulter, cat. no. A 63881) in accordance with the manufacturer's guidelines.

## Single-cell rt-qPCR
Primers targeting the *P. falciparum* 18S rRNA and the top 10 putative male and female driver genes (MDs and FDs) (Supplementary Table 2) were designed using the Dotmatic Geneious Prime software (Note: Genes PF3D7_0922300-FD10 (female) and PF3D7_1023600-MD9 (male) were excluded due to limitations in primer design). Primer sequences were subsequently blasted against the *P. falciparum* 3D7 reference genome to ensure specificity and prevent off-target amplification. In addition, primers for male (PF3D7_1477700 and PF3D7_1438800) and female (PF3D7_1447600 and PF3D7_1031000) sexual markers (Supplementary Table 2) from Ridgeway et al. were included[50]. All primers were procured from Integrated DNA Technologies (Coralville, Iowa, USA).

RT-qPCR experiments were performed using the POWER-SYBR Green chemistry, in a Quantstudio 5 (Applied Biosystems, Waltham, Massachusetts, EUA) with default conditions, including an initial denaturation step of 15 min at 95 °C, followed by 40 cycles of 15-s denaturation at 95 °C and 60-s annealing and extension at 60 °C. Nuclease-free water was employed as a negative control and all samples were tested in duplicates.

## rt-qPCR data analysis
The rt-qPCR data were analyzed following the 2−ΔΔCT methodology. All cycle threshold (CT) values were normalized to the reference gene (18S rRNA). The relative fold expression of single-cell and bulk samples was compared to the normalized expression of blood stages in bulk. To facilitate result interpretation, the natural base 10 logarithm of the fold expression difference was employed, to ensure results are presented on a comparable scale.

## Conservation analysis of the putative sex-driving genes
To assess whether the putative sex-driving genes were more conserved than other nuclear genes, we estimated selective pressures acting on these genes and compared them to a random set of nuclear genes. To estimate dN/dS per gene, the ratio of non-synonymous to synonymous substitution rates, we included our top 20 putative male and top 20 female driver genes (Supplementary Table 3) and extracted coding sequences of these genes from https://plasmodb.org/plasmo/app/ for *P. falciparum*. As a control, we also randomly selected 40 nuclear genes in this species as well as from the out group, *Plasmodium gaboni*, a closely related species. We used PlasmoDB to extract orthologs of the putative *P. falciparum* control sex-driving genes to align them with the sex-driving and control coding sequences from both *P. falciparum* and *P. gaboni*. Alignment quality was checked with the T-coffee alignment score[96], which was in the proximity of 90/100 for both control and sex-driving genes. We then ran condeml in PAML 4[97] with default options to estimate dN/dS per gene. To test for significant differences between gene sets, we used a Kruskal−Wallis test followed by a post hoc Dunn test in R v.2023.06.0 + 421. To control for the

impact of expression level on dN/dS, we specifically contrasted dN/dS values of the control and sex-driving genes with their expression values. To control for the impact of gene length on dN/dS, we reanalyzed genes shorter than or equal to 1700 base pairs.

## Reporting summary
Further information on research design is available in the Nature Portfolio Reporting Summary linked to this article.

## Data availability
The data generated for this study have been deposited in NCBI's Gene Expression Omnibus and are accessible through GEO Series accession number GSE226145 and interactively on [https://mubasher-mohammed.shinyapps.io/mohammedetal/]. The raw processed expression matrices, metadata, custom tracks (json), and *cis*-target motif databases are available via [https://zenodo.org/deposit/7652581]. We also provide a SCope session at [https://bit.ly/3KotDsP]. Publicly available datasets used for integration analyses include, bulk transcriptome data, van Biljon et al. (accession number GSE104889), single-cell transcriptome data from Gomes et al. can be found in the European Nucleotide Archive (accession code PRJEB48349).

## Code availability
The custom code scripts used to analyze data for this study are available at [https://github.com/ANKARKLEVLAB/Single-cell-P.f-gametocyte].

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

## Acknowledgements

We thank Cellbox (Hamburg, Germany) for support with the transport of live cells using their portable incubation system; Claudia Bergin and Anna-Maria Divne at the Microbial Single-Cell Genomics facility at Science for Life Laboratory (SciLifeLab), Sweden, for cell sorting and 10X Genomics applications; and the National Genomics Infrastructure, SciLifeLab for sequencing. The computations were performed using resources provided by SNIC through Uppsala Multidisciplinary Center for Advanced Computational Science (UPPMAX) under Project SNIC 2020/16-146 and SNIC 2021/22-492. This study is generously supported through grants from the Swedish Research Council (VR 2021-05057), and the Swedish Society for Medical Research (SSMF) to J.A.; the Swedish Research Council (VR 2021-06602) to J.H. and MIMS.

## Author contributions

J.A. conceived the study; J.A. and J.H. supervised the project; M.M., F.H., R.S., E.C., Y.V., and J.A. performed cell cultures and cell preparations, D.M. and E.C. performed the bulk rt-qPCR experiments; D.M. and J.A. performed the micromanipulation and single-cell RT-qPCR experiments; M.M. performed scRNA-seq data preprocessing and computational analysis, transcriptome assembly and interactive web visualization; Z.P. and T.S. performed the gene conservation analysis; J.A. and J.H. supervised the computational analyses; M.M., A.D., and J.A. coordinated the figure outline; M.M., A.D., D.M., Y.V., and J.A. wrote the manuscript, all authors provided input during the manuscript writing process.

## Funding

## Competing interests

The authors declare no competing interests.
