## [Peer Review File · Nature Communications]

Single-cell transcriptomics reveal transcriptional programs underlying male and female cell fate during *Plasmodium falciparum* gametocytogenesisEditorial Note: Parts of this Peer Review File have been redacted as indicated to remove third-party material where no permission to publish could be obtained.

REVIEWER COMMENTS

Reviewer #1 (Remarks to the Author):

Plasmodium falciparum displays a complicated lifecycle that transitions between a mosquito and a human host. While inside the human, the parasites replicate asexually, but transition to male or female gametocytes before being ingested by a mosquito. It is poorly understood how these parasites undergo sexual determination into male and female counterparts and may represent an important milestone in preventing transmission between humans and mosquitos. This study describes a single cell-based approach to characterize male and female gametocytes using a novel isolation approach that utilizes tdTomato expressing gametocytes. Strengths of the manuscript include the novel gametocyte isolation approach for scRNAseq studies, the quality of the scRNAseq data and its presentation, and the descriptive breadth of the study. Certainly, this dataset will be of value for researchers interested in studying gametocyte biology. However, this reviewer had difficulty with the lack of novel functional insight provided in the study. No genes were demonstrated functionally to play a role in gametogenesis and none of the insights from the scRNAseq data were validated by orthogonal approaches. Since at two previous studies captured the transcriptomes of gametocytes by scRNAseq (<https://doi.org/10.1038/s41467-021-23434-z> and DOI: 10.1126/science.aaw2619) it is not clear how much this work will push the needle in terms of advancing our understanding of *Plasmodium* sexual development.

Reviewer #2 (Remarks to the Author):

Review of Single-cell transcriptomics reveal transcriptional programs underlying male and female cell fate during *Plasmodium falciparum* gametocytogenesis

Mohammed et al performed scRNA-seq on developing gametocytes in *Plasmodium falciparum*. The authors identified clear trajectories of male and female development as well as expression patterns associated with each of these. They then go on to evaluate the splicing dynamics and velocities of genes that are potential drivers of the two cell fates, and confirm a subset of these using qPCR. Finally, the authors built regulatory networks by incorporating their expression data with binding motif data, revealing key transcription factors involved in gametocyte development.

I think this dataset is very useful to the malaria community and that the authors have done a very thorough job analyzing it.

Major comments

Generally, the text could be improved from removing redundancies and being more concise. I found there was a lot of explanation of pretty standard analysis methods in both the results and the methods.

More detail is needed on the induction method and corresponding time points both in figure 1 and results as well as the methods. The authors reference the Fivelman paper but don't

specify their exact induction conditions. For figure 1, it would be helpful to have more timing information displayed on 1A. Also, illustrate that the four tubes were pooled and that you have two replicates.

How did cell composition vary across the two replicates and how were the datasets integrated? Please include a UMAP colored by replicate or stacked bar chart that displays cells per cluster per replicate.

There is one cluster (C5) associated with Males and it looks disconnected from the early gametocyte clusters in the UMAP and PCA. I wonder if this is a numbers/sex ratio issue. If you downsample your female lineage to have equal representation of both sexes, how does that change your clustering and trajectory analysis?

Sup Fig 2B: Based on your comparison with the Biljon dataset, it seems like you have mostly later stage gametocytes or that your trajectories don't match real time changes. Could you please elaborate on this in the text? The comparison of the correlation plots between the bulk and single cell data 266-273 is not very informative. I would remove and replace with a more detailed analysis of how your clusters match cells in real time. Additionally, you could strengthen your staging by integrating with the single-cell data from Gomes et al 2023 that has both the WT and md-1 KO data.

Figure 5: Generally the figure and the associated text was difficult to digest. Could you expand on how this analysis relates to the genes that came up with the highest velocities in Figure 4, wouldn't those also qualify as driver genes? I think a clearer definition, both written and statistical, of driver gene is needed. How does your model allow for the inference of upstream factors in sex determination? Perhaps using md-1 as your control could allow you to validate the model. How did you select the genes for a more detailed analysis in D and E?

Minor comments

Line 4: There is no 5 address, so I assume Alexis Dziedziech should be 1,4.

Intro—line 47-55 That is a somewhat long intro on meiosis for a gametocyte dataset. I suggest cutting this and starting with line 55.

Line 72: LysoPC not LipoPC

Figure 1A—the last step in the diagram “single cell RNA sequencing” is not informative. Replace with information on how the sequencing was done.

Just a suggestion—I think the authors could combine figure 1 and 2 into one main figure and move 1D and 2B to the supplement. The UMAP gives a rough idea of the relationship between the clusters, and you have a clear separation of male and female clusters.

Sup Fig 1B—Why do C3 and C8 have such low n_umi and n_gene ? Are they committed rings or poor quality cells? This should be mentioned in the text.

Figure 3F: It is unclear to me why the males are included in this pseudotime branch. Shouldn't it be early gametocytes to mature females and a separate branch for the males?

Figure 5C: Include the total number assessed for each sex.

Line 958: Where was PAGA used?

Reviewer #3 (Remarks to the Author):

The author's performed single cell analysis of gametocytes harvested from days 3-7 and identified distinct population of gametocytes that were consistent with male and female gametocytes and then a third large population of multiple clusters that they classified as early gametocytes. The manuscript focused on the difference between males and females, not the classification of the early gametocyte populations. They develop a number of hypotheses about the genes and transcription factors driving the different profiles, but none of these hypotheses or genes were tested further experimentally. Consequently, the data is more of a discovery data set to be used in conjunction with all the prior work on the different expression profiles of males and female falciparum gametocyte, including a recent Nature article using single cell sequencing and identifying md1 as an important switch for sexual development between stage II and III gametocytes. Some mechanistic validation is needed to improve the impact.

Additionally, the different stages of the gametocytes observed at each time point should be indicated. Day 7 is quite early for mature gametocytes, was this intentional? Of particular interest is the % of mature stage IV & V gametocytes and whether the day 7 gametocyte cultures could be stimulated to formed gametes and exflagellate.

Was any cluster enriched in md1?

REVIEWER COMMENTS

General address:

Reviewer #1 (Remarks to the Author):

Plasmodium falciparum displays a complicated life cycle that transitions between a mosquito and a human host. While inside the human, the parasites replicate asexually, but transition to male or female gametocytes before being ingested by a mosquito. It is poorly understood how these parasites undergo sexual determination into male and female counterparts and may represent an important milestone in preventing transmission between humans and mosquitos. This study describes a single cell-based approach to characterize male and female gametocytes using a novel isolation approach that utilizes tdTomato expressing gametocytes. Strengths of the manuscript include the novel gametocyte isolation approach for scRNAseq studies, the quality of the scRNAseq data and its presentation, and the descriptive breadth of the study. Certainly, this dataset will be of value for researchers interested in studying gametocyte biology. However, this reviewer had difficulty with the lack of novel functional insight provided in the study. No genes were demonstrated functionally to play a role in gametogenesis and none of the insights from the scRNAseq data were validated by orthogonal approaches. Since at two previous studies captured the transcriptomes of gametocytes by scRNAseq (<https://doi.org/10.1038/s41467-021-23434-z> and DOI: 10.1126/science.aaw2619) it is not clear how much this work will push the needle in terms of advancing our understanding of *Plasmodium* sexual development.

We thank the reviewer for the comments and agree that our novel approach for enriching gametocytes or other parasite stages will be highly useful in further characterizing plasmodium stage development using high-throughput single cell omics. We also believe that this is the highest quality scRNA-seq dataset, specifically targeting *P. falciparum* gametocyte generated to date, which provides an important resource for the field.

Beyond the generation of the gametocyte single cell data, we believe that our computational approach is pushing the STOA in the malaria field. We have used a clear rationale in combining computational tools that are at the forefront of single-cell analyses, which enabled us to define the timing of expression and lineage-specificity of members from the AP2 family of transcription factors and establish motif-driven gene-regulatory co-expression networks. Combining pseudotime with RNA velocity helped us define panels of multiple putative driver genes for the male and the female cell fate, which provides important novel targets for the field, for future functional characterizations. Since RNA velocity relies on defining the presence or absence of introns, we validated these analyses using targeted qPCR analysis to define the presence or absence of the intron near the 3'-end of the transcript for a subset of the panel of putative male and female driver genes. We argue that the computational strategy implemented in our study will help guide future single cell studies in the field, with emphasis on studies that target cell differentiation and development.

We anticipate that our study as a whole, especially including the combination of computational analyses, and the optimization of these pipelines, have enabled us to uncover important novel information linked specifically to male and female gametocyte development, including, AP2 transcription factor lineage-specificity, as well as panels of novel putative genes that we propose to be involved in driving the male or female sexual cell fate.

Although we believe that functional validations of the panels of putative male and female driver genes is well beyond the scope of this study, we agree with the reviewer that additional orthogonal analyses will significantly strengthen our claims.

To this end, we have performed a set of orthogonal experiments to independently verify the timing of expression and to determine the male or female cell fate-specificity of nine of the top putative male and nine of the top putative female driver genes.

Experiment 1, in short:

Timing of expression of the putative M/F driver genes using rt-qPCR on bulk gametocyte RNA.

We used the *P. falciparum* NF54-Peg4 tdTomato line and experimentally induced gametocytes and subsequently harvested GlcNAc-treated cells in bulk at days; 2, 5 and 9 of gametocyte development, and performed RNA extraction. In order to further validate the timing of expression of putative M/F driver genes, we performed rt-qPCR, including a panel of newly designed primers covering the top nine putative male driver genes and the top nine putative female driver genes, together with previously published male gametocyte markers (PF3D7_1477700 and PF3D7_1438800) and female gametocyte markers (PF3D7_1447600 and PF3D7_1447600) (Ridgway, 2020 <https://doi.org/10.1128/mSphere.00671-20>), as well as primers for the 18S rRNA gene (PF3D7_0112300).

Data shown in **Supplementary Figure 5 A-C** and **Table S6**, and described on lines 489-502 in the results, lines 724-729 in the discussion and 1176-1189 in methods

Experiment 2, in short:

Cross-validation of sex-specific expression of the putative M/F driver genes using single-cell rt-qPCR.

We used the NF54-Peg4 tdTomato line, experimentally induced gametocytes and subsequently harvested cells at an early and a late time point during gametocyte development. tdTomato expression in combination with cell morphology were used to characterize developing gametocytes in an inverted, fluorescence microscope. Individual gametocytes were isolated using capillary-based micromanipulation (Eppendorf) in a Leica DMI8 inverted, wide-field, fluorescence microscope, and deposited in smart-seq2 lysis buffer in individual microcentrifuge tubes. The smart-seq 2 chemistry was applied (Mohammed et al, Microbiology Spectrum, 2023 - *doi: 10.1128/spectrum.03671-22*), including; cell lysis, cDNA synthesis and strand-switching, followed by amplification over 24 cycles, and cDNA purification. The samples were subsequently analyzed using rt-qPCR, including a panel of newly designed primers covering the top nine putative male driver genes and the top nine putative female driver gene (same as in the section above), together with previously published male gametocyte markers (PF3D7_1477700 and PF3D7_1438800) and female gametocyte markers (PF3D7_1447600 and PF3D7_1447600) (Ridgway, 2020 <https://doi.org/10.1128/mSphere.00671-20>), as well as primers for the 18S rRNA gene (PF3D7_0112300).

Data shown in **Supplementary Figure 5 D and E** and **Table S6**, and described on lines 503-530 in the results, lines 729-750 in the discussion and 1191-1233 in methods

In addition, we have performed computational analyses to assess whether the putative male and female (M/F) driver genes were more conserved than other nuclear genes, we estimated selective pressures acting on these genes and compared them to a random set of nuclear genes. The ratio of nonsynonymous to synonymous substitution rates (d_N/d_S) is often used to assess selection based on divergence data. To estimate d_N/d_S per gene, we extracted putative coding sequences of sex-driving genes from <https://plasmodb.org/plasmo/app/> for *P. falciparum*. As a control set of genes we also randomly selected 40 nuclear genes in this species. *P. gaboni*, a closely related species, was used as an outgroup and we also used PlasmoDB to extract orthologs of the control and the putative M/F driver genes in this species. We aligned coding sequences of both species for the putative M/F driver genes and control genes using T-coffee (Notredame et al. 2000, J. Mol. Biol.). Alignment quality was checked with the T-coffee alignment score and was around 90/100 for both the control and the putative M/F driver genes. We then ran condemn - PAML 4 (Yang 2007, Mol. Biol. Evol.) with default

options to estimate d_N/d_S per gene. To test for significant differences between sets of genes, we used a Kruskal-Wallis test followed by a post-hoc Dunn test in R v.2023.06.0+421. To control for the impact of gene length on d_N/d_S , we reanalyzed only genes shorter than or equal to 1700 base

Data shown in **Supplementary Figure 6**, and described on lines 532-558 results, lines 752-762 in discussion and lines 1234-1250 in methods.

We agree with the reviewer that the referenced single cell studies have contributed majorly to the field. However, regarding the referenced single cell studies by; Real et al, Nat Com, 2021 - *P. falciparum* development in the mosquito, see **Rev Figure 1**, and Howick et al, Science, 2019 - single cell atlas of several developmental stages including both *P. berghei* and *P. falciparum*, see **Rev Figure 2**, they only include a small minority of cells covering the *P. falciparum* gametocyte stage.

It is noteworthy that the target of these studies was not the gametocyte developmental stage and both studies only included a small proportion of gametocytes from *Plasmodium falciparum*. In our study, the aim was to provide a cohesive, single cell transcriptome map of *P. falciparum* development from the very early developing gametocyte and through development to mature gametocytes, exclusively, including the developmental trajectories of both male and female gametocytes.

[Redacted]

Rev Figure 1. Real et al, Nat Com, 2021, highlighting the major focal point of this study - characterization of *Plasmodium* development in the mosquito vector

[Redacted]

Rev Figure 2. Howick et al, Science 2019, a single cell atlas of the *Plasmodium* lifecycle, but focusing on the rodent-infectious strain, *P. berghei*, and with very limited numbers of *P. falciparum* gametocytes, as indicated in the figure.

Reviewer #2 (Remarks to the Author):

Review of Single-cell transcriptomics reveal transcriptional programs underlying male and female cell fate during Plasmodium falciparum gametocytogenesis

Mohammed et al performed scRNA-seq on developing gametocytes in Plasmodium falciparum. The authors identified clear trajectories of male and female development as well as expression patterns associated with each of these. They then go on to evaluate the splicing dynamics and velocities of genes that are potential drivers of the two cell fates, and confirm a subset of these using qPCR. Finally, the authors built regulatory networks by incorporating their expression data with binding motif data, revealing key transcription factors involved in gametocyte development.

I think this dataset is very useful to the malaria community and that the authors have done a very thorough job analyzing it.

Major comments

Generally, the text could be improved from removing redundancies and being more concise. I found there was a lot of explanation of pretty standard analysis methods in both the results and the methods.

We thank the reviewer for this observation. Explanations about methods and redundancies were removed from several sections in the results. For example, lines: 55-58 ff, 165-167 ff., 204-209 ff, 218-220 ff, 224-227 ff, 332-333 ff, 384-392 ff and 1003 ff

More detail is needed on the induction method and corresponding time points both in **Figure 1** and results as well as the methods. The authors reference the Fivelman paper but don't specify their exact induction conditions. For **Figure 1**, it would be helpful to have more timing information displayed on **1A**. Also, illustrate that the four tubes were pooled and that you have two replicates.

We thank the reviewer for these suggestions. We have updated the methods to include induction details, lines: 971-977. We updated the schematic in **Figure 1A** to include the four time points of gametocyte enrichment, the pooling of gametocytes and the subsequent loading of the cells into the 10X Chromium controller for scRNA sequencing, lines: 979 - 985.

How did cell composition vary across the two replicates and how were the datasets integrated? Please include a UMAP colored by replicate or stacked bar chart that displays cells per cluster per replicate.

UMAPs showing the overlap between the two replicates has now been included as a new Supplementary Fig 1F (left: original UMAP and right: UMAP including batch correction). We have qualitatively compared our analysis with and without batch correction using BBKNN and have not seen any large deviations in the results. Since the batch corrected UMAP does not seem to mix the cells better (**Supplementary Figure 1F**), we chose to not apply batch correction in the final analysis. This is because batch correction performs assumptions that can also add bias, and should only be performed if deemed necessary. The proportions of each library, for each cluster, is seen in new **Supplementary Figure 1G**.

Descriptions of these analyses have been added on lines 1035 -1040 in methods

There is one cluster (C5) associated with Males and it looks disconnected from the early gametocyte clusters in the UMAP and PCA. I wonder if this is a numbers/sex ratio issue. If you

downsample your female lineage to have equal representation of both sexes, how does that change your clustering and trajectory analysis?

We thank the reviewer for this suggestion. We have performed random down-sampling while keeping the feature selection of highly variable genes by subtracting random cell barcodes from the previously annotated female lineage in order to provide unbiased representation of cells for subsequent analysis including the reclustering (**Rev Figure S1**). Our analysis (Rev Figure S2) indicates that the down-sampling doesn't affect the clustering of the distinct male lineage population. We observe a nearly identical transcriptionally divergent PCA (Rev Figure S1B:Top panel) as compared to the initial analysis (Rev Figure S1A:Top panel). Further, by mapping male and female cells from the original analysis, we observed that the transcriptional patterns remain largely consistent. We expanded our investigation to assess the impact of varying clustering resolutions on the down-sampled cells, conducting tests at resolutions of 0.25, 0.5, and 1 to determine their effect on the clustering of both male and female populations (**Rev Figure S1C**). By employing three distinct resolutions across a wide spectrum of community detection, we aimed to capture any shifts in the ephemeral cell states of both the male and female lineages. Our findings indicate negligible differences in the transcriptionally unique male lineage between the 0.25 and 0.5 resolutions. However, as we moved to a higher resolution, an increase in cell populations was noted, particularly within the early gametocyte and female lineage. At a resolution of 1, we identify specific cell states within the male cluster (C5), illustrating the transcriptional diversity between the early and late stages of the male lineage, while maintaining a distinct, overall transcriptional identity characteristic of the male lineage.

Next we performed trajectory analysis on the down-sampled data (**Rev Figure S2**). Here we can still observe lineage divergence when ordering the cells along the pseudotime trajectory.

We believe that the suggested analysis has provided a good validation of our findings, but we do not see a reason to include the down-sampling analysis in the main or supplementary materials of the manuscript. We would however like to seek the reviewer's advice on this.

Rev Figure S1

A

Before downsampling female lineage

Total cells = 4555
 Early gametocyte = 1939
 Female lineage = 2148
 Male lineage = 468

B

After downsampling female lineage

Total cells = 2875
 Early gametocyte = 1939
 Female lineage = 468
 Male lineage = 468

C

Re-clustering with 3 resolutions
 after downsampling female lineage

Rev Figure S2

Trajectory analysis after downsampling the female lineage

Total cells = 2875
Early gametocyte = 1939
Female lineage = 468
Male lineage = 468

Sup Fig 2B: Based on your comparison with the Biljon dataset, it seems like you have mostly later stage gametocytes or that your trajectories don't match real time changes. Could you please elaborate on this in the text?

We thank the reviewer for this comment. First off, since our data represents the expression of individual gametocytes, we argue that it resembles real time changes at much higher resolution than does a bulk dataset. The van Biljon dataset is undoubtedly of high importance to the malaria field, but since an originally synchronous culture of cells tends to vary in their development (i.e. synchronicity) over time, the bulk dataset is destined to represent gametocytes of varying developmental stages and also a "pooled" expression of male and female genes in mid to late timepoints. Thus a comparison between our single cell data and the bulk data from the van Biljon study is not completely straight fwd.

That said, we have reanalyzed the correlation between our scRNA-seq data with the bulk RNA-seq data from the van Biljon et al study (Rev Figure S3). In this analysis we followed a deconvolution linear regression model to calculate the cell type proportion using the shared expression profile between the bulk RNA-seq data and our single cell data. We then fit the regression model to transfer the cell type labels (**stage and time point identifiers from the van Biljon bulk dataset**) onto our single cell data and aligned the transferred labels on our pre-estimated pseudotime trajectory. Our analysis shows that the bulk labels align with our pseudotime data for these stages (Rev Figure S3A). Note that our bulk deconvolution analysis initially included all the bulk transcriptome data points from the van Biljon study, however, we were not able to achieve optimal "label transfer" due to the variability of the transcriptome in the early and late collected stages in the bulk data. This was similarly described by Gomes and colleagues (Gomes et al, Nature, 2022) in their study calculating the expression correlation between the bulk and single cell data. Upon several iterations of the deconvolution analysis and considering the noise introduced by the bulk RNA-seq data, we imputed the bulk data to remove outliers from each time point or stage. We subset the bulk data from day 3 to day 7. Our pseudo-bulk alignment showed the relative clustering between our single cell data and the bulk RNA-seq (**Rev Figure S3A**). We further wanted to investigate the cell type allocations within the annotations of male and female lineages. The data showed that the male lineage differentiation still occurs at or prior to what is referred to as day 4 - stage I to II in the Biljon study (Rev Figure S3B right panel) with the majority of male lineage cells appearing to express a high number of mature male transcripts around, what is referred to as day 6 in the Biljon study (**Rev Figure S3B**).

We have made additions to the text on lines: 264-268

Rev Figure S3

The comparison of the correlation plots between the bulk and single cell data 266-273 is not very informative. I would remove and replace with a more detailed analysis of how your clusters match cells in real time.

We agree with the reviewer and have performed the analyses included in Rev Figure S3, which we anticipate is a more detailed analysis of how our clusters match with the van Biljon bulk dataset. See description above. If the reviewer agrees that this is a clearer way of showing the comparison, we will be happy to add to or replace the previous figure.

Additionally, you could strengthen your staging by integrating with the single-cell data from Gomes et al 2023 that has both the WT and md-1 KO data.

We have followed the reviewer's suggestion and integrated our data set with both WT and KO single cell data from Gomes et al. data set (Rev Figure S4A). We were able to observe a clear bifurcation trajectory that aligned well with the Gomes et al dataset. We then compare the number of targeted sexual stages across the datasets. Our data indicates the following ratio of cell types: progenitor cells = 42.6%, female lineage = 47.2% and male lineage = 10.3%, in comparison with the Gomes et al data, which indicates the following ratio of cell types: progenitor cells = 36%, female lineage = 58.7% and male lineage 5.3% (Rev Figure S4D).

We integrated our data set with the Gomes et al WT data set as shown in (Rev Figure S4B). We used a gene expression matrix correction method (Scanorama v1.7.4) that relies on correction of the sequencing depth to obtain a scalable gene expression profile across the single cells from different datasets. We then calculated the top 2000 highly variable genes and unified the variable genes to subsequently calculate the dimensionality reduction for the Gomes et al and Mohammed et al integrated datasets and show that there is overlap between the two datasets (Rev Figure S4B:Left panel). Similar alignments of progenitor, male and female clusters were evident when overlaying the Gomes et al WT dataset with the Mohammed et al data, or vice versa(Rev FigureS4:Middle panel).

We then integrated our dataset with the Gomes et al KO data set (Rev Figure S4C). Here we observed, as expected, that no male gametocytes were present, which validates the male lineage found in our study (Rev Figure S4C middle panel).

The two single cell datasets, as such, are complementary, however, the aims, execution and outcomes are vastly different, where our aim has been to uncover multiple genes and genetic regulators involved in *P.falciparum* sexual cell fate in both the male and female lineages.

Rev Figure S4

Integration: Wild type Gomes et al & Mohammed et al

Integration: KO Gomes et al & Mohammed et al

Figure 5: Generally the figure and the associated text was difficult to digest. Could you expand on how this analysis relates to the genes that came up with the highest velocities in Figure 4, wouldn't those also qualify as driver genes? I think a clearer definition, both written and statistical, of driver gene is needed.

We thank the reviewer for addressing this. For the putative driver genes (**Figure 5B**, now **Figure 4B**), we model the correlation of the expression of the genes to either male or female mature cells, the genes are then ranked based on their expression correlation from high to low correlation towards either C5 (male cells) or C1 (female cells). To perform this analysis, we combined the splicing kinetics data from the initial mapping of the raw sequencing data in a loom file. We then added the processed gene expression matrix with cell annotations and used predefined male and female gametocyte markers to create a model in order to infer the expression correlation trends for the genes in our processed data set. The correlation results for the expression of genes were then ranked based on the significance.

In (**Figure 4D**, now **Figure 3D**) we measured the ratio between pre-mature mRNA and mature mRNA for the top 5 velocity genes in C5 and C1 exclusively, we did not include the splicing ratio for top ranked genes in other clusters. This analysis was performed in order to indicate which genes were expressed the most in late stage males and females populations. However, these genes did not qualify as putative driver genes since we did not observe a continuous expression correlation with high correlation significance during modeling of the expression we used in **Figure 4B**.

We added an explanation in the manuscript on lines 359 - 366 and 399 - 404 in the results section

How does your model allow for the inference of upstream factors in sex determination? Perhaps using md-1 as your control could allow you to validate the model. How did you select the genes for a more detailed analysis in D and E?

In our study, RNA velocity was employed to analyze immature mRNAs of specific genes. Combined with pseudotime analysis, this method allowed us to track the expression of these genes throughout the development of the male and female lineages, shedding light on potential upstream factors in sex determination. However, it is important to note that our single-cell RNA sequencing (scRNA-seq) dataset has limitations and doesn't provide insights into regulatory factors influencing gene expression, for example, through integration of a proteomic or epigenomic dataset. Consequently, we selected genes with sufficient 3'-end coverage (around 300pb), considering the 3' bias in our single cell dataset due to the nature of the 10X Genomics chemistry and discriminated for genes with exon-intron junctions covering at least several bases. This selection criterion, which includes the estimation of the ratio between spliced and unspliced reads, guided our investigation in verifying the chosen putative driver genes.

Minor comments

Line 4: There is no 5 address, so I assume Alexis Dziejch should be 1,4.

The address has been updated

Intro—line 47-55 That is a somewhat long intro on meiosis for a gametocyte dataset. I suggest cutting this and starting with line 55.

The meiosis section has been cut down

Line 72: LysoPC not LipoPC

This has been corrected

Figure 1A—the last step in the diagram “single cell RNA sequencing” is not informative. Replace with information on how the sequencing was done.

This has been changed accordingly (see new **Figure 1**)

Just a suggestion—I think the authors could combine figure 1 and 2 into one main figure and move 1D and 2B to the supplement. The UMAP gives a rough idea of the relationship between the clusters, and you have a clear separation of male and female clusters.

We thank the reviewer for this suggestion, Figures 1 and 2 have been combined to a new **Figure 1** and subfigures from previous Figure 1 and Figure 2 that are no longer included in the main figures have been moved to **Supplementary Figure 1**.

Sup Fig 1B—Why do C3 and C8 have such low n_umi and n_gene? Are they committed rings or poor quality cells? This should be mentioned in the text.

We tested for the number of genes and UMIs for each of the clusters. We observed variability in the number of genes and transcripts assigned to each cluster, including, C3 and C8. Cells that make up clusters C3 and C8 passed our cut off of a minimum of 200 genes expressed per single cell during the initial data preprocessing. During data preprocessing, we decided to set the cut off at 200 n_genes in order not to exclude potential gametocyte ring stage parasites, due to their generally low global expression of genes. We provide (**Rev1_Figure S5**) for a more thorough examination of these clusters including testing for early genes provided by Silvestirini et al 2010 PMID: [20332084](https://pubmed.ncbi.nlm.nih.gov/20332084/). We selected 5 representative genes from the gene list in Table 2 in the article, including: ('PF3D7_1253000', 'PF3D7_1102500', 'PF3D7_1429600', 'PF3D7_0936600' and 'PF3D7_0114000'). We observed that the C3 cluster expressed the selected early genes, shown here in (**Rev1_Figure S5B**). The localization of C3 among early gametocytes along with the expression of the early markers indicates that C3 is an early gametocyte cell population

Expression of the same markers are not, with minor exceptions, present in the C8 cluster (**Rev1_Figure S5C**), as this is a mature stage population. Based on the expression of several mature female gametocyte markers, we can define that C8 is a mature female subpopulation, but it is difficult to define if the low n_umi and n_gene is low due to poor quality cells, technicalities during library preparations or due to biological reasons, if for example C8 represents a transient state of development.

We have now made a short addition to address this on lines 1078-1080

Rev Figure S5

A

B

C

Figure 3F: It is unclear to me why the males are included in this pseudotime branch. Shouldn't it be early gametocytes to mature females and a separate branch for the males?

Our aim was to concurrently illustrate the differential expression of IMCs in both male and female populations. To achieve this, we utilized a pseudotime heatmap for visualization. It's important to note that isolating the male population for this analysis could potentially alter the pseudotime units. To ensure a balanced representation, we standardized these units across the entire dataset, thereby preventing any inadvertent skewing of results towards specific cell populations. This approach was integral to maintaining the integrity and unbiased nature of our analysis.

Figure 5C: Include the total number assessed for each sex.

This information has been added to the new **Figure 4C**

Line 958: Where was PAGA used?

We used the PAGA graph abstraction as dependency of the Cellrank package pipeline to infer the terminal state probability for C5 and C1 in **Supplementary Figure 3B**.

Added in **Supplementary Figure 3** legend on line: 1460

Reviewer #3 (Remarks to the Author):

The author's performed single cell analysis of gametocytes harvested from days 3-7 and identified distinct population of gametocytes that were consistent with male and female gametocytes and then a third large population of multiple clusters that they classified as early gametocytes. The manuscript focused on the difference between males and females, not the classification of the early gametocyte populations. They develop a number of hypotheses about the genes and transcription factors driving the different profiles, but none of these hypotheses or genes were tested further experimentally. Consequently, the data is more of a discovery data set to be used in conjunction with all the prior work on the different expression profiles of males and female falciparum gametocyte, including a recent Nature article using single cell sequencing and identifying md1 as an important switch for sexual development between stage II and III gametocytes. Some mechanistic validation is needed to improve the impact.

Additionally, the different stages of the gametocytes observed at each time point should be indicated. Day 7 is quite early for mature gametocytes, was this intentional? Of particular interest is the % of mature stage IV & V gametocytes and whether the day 7 gametocyte cultures could be stimulated to form gametes and exflagellate.

Was any cluster enriched in md1?

We thank the reviewer for their comments. We also contend that our scRNA-seq dataset, which specifically focuses on *P. falciparum* gametocytes, stands as the most comprehensive resource of its kind to date, offering significant value to the scientific community. Kindly note that gametocytes from the NF54 tdTomato line were harvested and enriched at days: 1 through 7 (not 3-7 as indicated in the reviewer's description). In our hands, using the described cell line, we do have a combination of stages ~III-V on day 7. The aim for our analyses was to trace development from the very early stages of gametocyte development leading up to mature

male and female gametocytes, but since mature (Stage V) male and female gametocytes have previously been analyzed we didn't put major emphasis of generating an abundance of that stage. In Figure 1E, we overlay the UMAP with several mature male and female markers from Walzer et al, mSphere, 2018, indicating that we have large defined clusters of mature male and female gametocytes among our analyzed single cells.

Regarding prior single-cell studies such as those by Real et al., in Nat Communications 2021, which investigated *P. falciparum*'s development within mosquitoes, and Howick et al., in Science 2019, which presented a single-cell atlas of various developmental stages of both *P. berghei* and *P. falciparum* including a minor focus on *P. falciparum* gametocytes, it's important to highlight that these studies did not primarily focus on the gametocyte stage and included only a small fraction of gametocytes.

In reference to the observations made in the Gomes study regarding the gene *md1*, it was noted that *md1* expression is not limited to male-only expression but occurs broadly. This lack of specificity to either male or female developmental pathways in our context means *md1* cannot serve as a reliable control for our analysis. The distinct nature of the research questions our study seeks to answer necessitates markers with clear lineage-specific expression patterns. Given *md1*'s general expression profile, it does not meet the criteria for a control in the differential analyses we conducted, where specificity is crucial for interpreting the biological significance of our findings.

In contrast, our research aimed to meticulously chart the development of *P. falciparum* from the very early gametocyte stages to maturity, focusing exclusively on mapping the transcriptome of both male and female gametocytes and their developmental pathways.

Furthermore, we assert that our computational methodology is advancing the state of the art within the malaria research domain. By integrating leading-edge computational tools for single-cell analysis, we have pinpointed the expression timing and lineage specificity of AP2-family transcription factors and delineated motif-based gene-regulatory networks. The integration of pseudotime analysis with RNA velocity has enabled us to identify potential key driver genes for both male and female gametocyte development, presenting new targets for research.

Our computational approach, particularly through the use of pseudotime in combination with RNA velocity, was further validated by targeted qPCR to confirm the presence or absence of specific introns, enhancing our understanding of male and female driver genes. We propose that our computational strategy will serve as a valuable framework for future studies focusing on cellular differentiation and development.

Overall, our study not only elucidates novel insights into the development of male and female gametocytes but also highlights a suite of potential genes implicated in sex-specific cell fate determination. While functional validation of these gene panels falls outside this study's scope, we acknowledge the reviewer's suggestion that further orthogonal analyses would substantiate our findings.

To this end, we conducted orthogonal experiments to independently confirm the expression timing and determine the specificity of cell fate for the top candidate driver genes for both sexes. Briefly, we utilized the NF54-Peg4 tdTomato line to induce gametocyte development, isolated gametocytes at key developmental stages, and performed smart-seq2 analysis followed by rt-qPCR targeting both newly identified and previously recognized gametocyte markers (see **Supplementary Figure 5**). (Further described in the response to Reviewer 1)

The results, depicted in **Supplementary Figure 6**, not only validate our findings but also contribute to our understanding of the evolutionary pressures acting on these putative driver genes compared to a random selection of nuclear genes, using dN/dS ratios as a measure of

selection. This comprehensive analysis, underscores the level of conservation and evolutionary significance of the identified male and female driver genes, providing a solid foundation for future research into *Plasmodium* gametocyte development.

Experiment 1, in short:

Timing of expression of the putative M/F driver genes using rt-qPCR on bulk gametocyte RNA.

We used the *P. falciparum* NF54-Peg4 tdTomato line and experimentally induced gametocytes and subsequently harvested GlcNAc-treated cells in bulk at days; 2, 5 and 9 of gametocyte development, and performed RNA extraction. In order to further validate the timing of expression of putative M/F driver genes, we performed rt-qPCR, including a panel of newly designed primers covering the top nine putative male driver genes and the top nine putative female driver gene, together with previously published male gametocyte markers (PF3D7_1477700 and PF3D7_1438800) and female gametocyte markers (PF3D7_1447600 and PF3D7_1447600), as well as primers for the 18S rRNA gene (PF3D7_0112300).

Data shown in **Supplementary Figure 5 A-C** and **Table S6**, and described on lines 489-502 in the results, lines 724-729 in the discussion and 1176-1189 in methods

Experiment 2, in short:

Cross-validation of sex-specific expression of the putative M/F driver genes using single-cell rt-qPCR.

We used the NF54-Peg4 tdTomato line, experimentally induced gametocytes and subsequently harvested cells at an early and a late time point during gametocyte development. tdTomato expression in combination with cell morphology were used to characterize developing gametocytes in an inverted, fluorescence microscope. Individual gametocytes were isolated using capillary-based micromanipulation (Eppendorf) in a Leica DMI8 inverted, wide-field, fluorescence microscope, and deposited in smart-seq2 lysis buffer in individual microcentrifuge tubes. The smart-seq 2 chemistry was applied (Mohammed et al, Microbiology Spectrum, 2023 - doi: 10.1128/spectrum.03671-22), including; cell lysis, cDNA synthesis and strand-switching, followed by amplification over 24 cycles, and cDNA purification. The samples were subsequently analyzed using rt-qPCR, including a panel of newly designed primers covering the top nine putative male driver genes and the top nine putative female driver gene (same as in the section above), together with previously published male gametocyte markers (PF3D7_1477700 and PF3D7_1438800) and female gametocyte markers (PF3D7_1447600 and PF3D7_1447600), as well as primers for the 18S rRNA gene (PF3D7_0112300).

Data shown in **Supplementary Figure 5 D and E** and **Table S6**, and described on lines 503-530 in the results, lines 729-750 in the discussion and 1191-1233 in methods

In addition, we have performed computational analyses to assess whether the putative male and female (M/F) driver genes were more conserved than other nuclear genes, we estimated selective pressures acting on these genes and compared them to a random set of nuclear genes. The ratio of nonsynonymous to synonymous substitution rates (d_N/d_S) is often used to assess selection based on divergence data. To estimate d_N/d_S per gene, we extracted putative coding sequences of sex-driving genes from <https://plasmodb.org/plasmo/app/> for *P. falciparum*. As a control set of genes we also randomly selected 40 nuclear genes in this species. *P. gaboni*, a closely related species, was used as an outgroup and we also used PlasmoDB to extract orthologs of the control and the putative M/F driver genes in this species. We aligned coding sequences of both species for the putative M/F driver genes and control genes using T-coffee (Notredame et al. 2000, J. Mol. Biol.). Alignment quality was checked with the T-coffee alignment score and was around 90/100 for both the control and the putative M/F driver genes. We then ran condemn - PAML 4 (Yang 2007, Mol. Biol. Evol.) with default

options to estimate d_N/d_S per gene. To test for significant differences between sets of genes, we used a Kruskal-Wallis test followed by a post-hoc Dunn test in R v.2023.06.0+421. To control for the impact of gene length on d_N/d_S , we reanalysed only genes shorter than or equal to 1700 base

Data shown in **Supplementary Figure 6**, and described on lines 532-558 results, lines 752-762 in discussion and lines 1234-1250 in methods.

Was any cluster enriched in md1?

We analyzed the expression of md1 across the clusters (**Rev Figure S6 A**) and found that there is a higher abundance of md1 transcripts in C8, which we have identified as a mature female cluster. In addition, there appears to be a slightly higher abundance of md1 transcripts in C3 and C6, both of which are identified as early developing gametocytes.

Using the Scanpy function (`scanpy.tl.rank_genes_group`) to rank genes in C8 based on the Wilcoxon-rank-sum test. This analysis is based on comparing top rank genes in C8 to the rest of the clusters as shown in **Rev Figure S6 B**. we found that md-1 (PF3D7_1438800) is ranked among the top 20 ranked genes of C8 (number 15). md-1 showed 2.2 log fold change expression with p value < 0.01.

Rev Figure S6

REVIEWERS' COMMENTS

Reviewer #1 (Remarks to the Author):

The authors appear to have addressed all of my and the other reviewers concerns. This remains an important piece of work and I look forward to seeing it in press.

Reviewer #3 (Remarks to the Author):

The authors have added two experiments to further evaluate the expression profile of the driver genes they identified and in the response to the reviewers demonstrated the concordance of their data with Gomes et al. The new data is a valuable addition and confirmed the predicted expression profile of the female driver genes and some of the male driver genes. The new data also clearly demonstrates the discovery nature of the data set. It is a useful tool toward dissecting male and female gametocyte development. However, as they state in the final paragraph further work is needed to experimentally confirm the functional and regulatory roles of the highlighted genes and lineages. The complicated pattern of the clusters designated early gametocytes could also benefit from additional focused evaluation of these stages. Consequently, throughout the manuscript terms like validated, regulate and mechanism need to be replaced or qualified. I have attempted to identify the sentences that need to be revised, but might have missed some that also need to be changed.

Line 136 Clarify that the IMC is also produced in males (as discussed in line 431) and females, although differences were found in the expression of these associated genes.

Line 178 Please revise or remove the end of the sentence. The appearance of ETRAMP RNA in clusters C0, C3, C4 and C9 says nothing about protein transport, it just means ETRAMP RNA is expressed in cells present in clusters that are proposed to be early gametocytes.

213-225 Most of the gene listed here are not shown in Fig 2E. What are the profiles of the genes mentioned in the text, especially those that are mentioned as previously identified with early gametocytes. EXP1 (PF3D7_1121600), P113 (PF3D7_1420700), ETRAMP4 (PF3D7_0423700) and HDP1 (PF3D7_1466200)

233 (Supplementary figure 1D, bottom panel). Change "validate" to "support". The results are from in silico analysis.

382-384 This sentence needs to be removed or extensively revised. No regulatory function has been shown for any of these genes or their splicing. The findings may suggest something that could be further tested, but by themselves they do not add significantly to our knowledge about the regulatory activity of transcription.

384-387 The different patterns of unspliced RNA in males and females are consistent with Gomes et al.'s finding but they do not directly support their data on the role of md1 splicing. To support their findings, you would have had to demonstrate differential splicing of md1 in your male and female populations, however this was not assessed and this sentence needs to be revised

422 I think “recover” should be replaced with “reveal”

431 Please clarify how this relates to the low RNA levels for many of the IMC associated genes shown in Figure 1G&H.

441-446 This section should be moved to the discussion and lines 441-443 tempered to reflect that it is all based on RNA sequencing. No actual experiments have been done evaluating the regulation of gene activity programs or the role of splicing kinetics in influencing sex transcriptional changes. This requires experimental modulation of the parameter to demonstrate the effect.

453-455 Please revise this sentence. The RT-qPCR data used to confirm the expression of the putative driver genes does not clearly demonstrate that there is lower expression at early time points for the 9 male genes, MD5 was not detected and MD4 & MD8 decline. From the individual cell sequencing presented later, there is also little difference between the expression of the driver genes and the two time points (early to mid and mid to late) and one of the male drivers is also expressed in some female gametocyte.

In Supplementary fig S5B&C The results from the individual genes at the 3 time points should be clustered together, not separated by time point, to allow direct analysis of the trend for each gene over time,

469 the same gene is listed twice.

495 Does the analysis in Supplementary figure S6 use all the 9 male driver genes even those that did not work well in the RT-qPCR analysis?

514 Please remove or revise the end of the sentence. No evidence is provided that these genes have a functional role in the sexual determination mechanisms. Some of them could be markers to define the lineage, but this does not imply a regulatory relationship. None of the driver genes in Table 6 have any annotated or experimental role in gene regulation.

532-541 Move this to the discussion.

600-602 Move this to the discussion.

603 Add “ is predicted to” before regulates

628 Add “predicted” before “repressor”

648 “ensure validation” is too strong please replace with the following or similar: We used previously reported canonical markers of mature gametocytes to confirm the mapping of differentiated mature male and female gametocyte cell populations in our single-cell dataset

655 add “selected” before “putative driver genes”. Not all 100 driver genes were tested.

675 Please rephrase this. The results confirm an increase in expression over time for the female genes tested, but not the male genes (Supplementary Fig 5A,B&E).

680 This needs to be qualified in that a number of male driver genes were not consistently or

selectively expressed in the males Supplementary Fig 5E. There was also no difference in gene expression between individually captured early-mid and mid-late gametocytes in Supplementary Fig 5E.

688-692 Both sentences should be removed. No evidence is presented for MD7 to be essential in males, not females. Nor is any data provided that most lineage markers are essential for development. The profile for MD7 is quite different from either MM1 or MM2. I assume MM2 is PF3D7_1438800, md1 of Gomes et al.. The definition of the MMs and FMs should be given in the figure legend for Supplementary Fig 5.

705-706 Please remove this phrase. The male and female driver genes have a dN/dS significantly above nuclear genes and no other evidence is presented supporting the statement that they are essential for parasite fitness and survival.

709-726 Could be removed or significantly shortened. The differential expression of a gene in males and females is not a new insight into the potential role of TRF1 in gametogenesis.

755-761 It should be noted that the complex mix of clusters designated as early gametocytes suggests that more work is needed to define the early steps of male and female gametocyte production.

769 Change “validates” to “supports”

791 Add “that its expression profile suggests it” before “regulates at least two genes, of which one (PF3D7_0209500) has not yet been functionally annotated” No evidence is presented that AP2-G2 actually regulates these two genes.

826 Please revise this sentence to clarify its meaning.

831 Replace “validate” with “identify”. Validation implies confirming a functional link which was not done.

841 Change “indicating” to “suggesting”

860-878 This section is unrelated to the work reported in this study and should be deleted. Their study examines sexual development, not sexual conversion or epigenetic regulation. Sexual conversion is initiated during the asexual cycle, which disrupted by the application of NAG.

881 add “suggests” before “gene regulatory” and replace “mechanisms” with pathways.

Figure legends: Please define the number of cells used as the denominator for the prevalence of the genes listed in the all the dot plots.

Reviewer #4

Single-cell transcriptomics reveal transcriptional programs underlying male and female cell fate during *Plasmodium falciparum* gametocytogenesis (revised manuscript)

In this study by Mohammed et al., the authors performed single-cell RNA seq on *P. falciparum* gametocytes to further characterise mechanisms of sex determination. They identify different transcriptional programs associated with male and female developmental, as well as putative driver genes for the male and female trajectories. Additionally, the authors perform motif-driven gene regulatory network analysis that allows them to predict how sex discrimination is regulated. As described by the previous Reviewer 2, this is a large and very comprehensively analysed data set that significantly contributes to our understanding of male and female gametocyte development in *P. falciparum*. The revised manuscript is much improved; however, some issues raised by the previous Review 2 remain outstanding.

Major comments:

Generally, the text could be improved from removing redundancies and being more concise. I found there was a lot of explanation of pretty standard analysis methods in both the results and the methods.

The text remains very long and would benefit significantly from additional trimming/editing for conciseness. The discussion section in particular repeats many of findings described in the results and should be more tightly focused on analysing the implications of the data.

More detail is needed on the induction method and corresponding time points both in Figure 1 and results as well as the methods. The authors reference the Fivelman paper but don't specify their exact induction conditions. For Figure 1, it would be helpful to have more timing information displayed on 1A. Also, illustrate that the four tubes were pooled and that you have two replicates.

Some additional information would still be warranted here. It would be helpful to indicate somewhere in the methods that multiple cultures were maintained in parallel so that all stages could be collected on the same day (if this was the case). In the figure (panel 1A), the shaded arrow with FACS written underneath is confusing and suggests samples are collected on days -3 to -1. To further clarify induction method, it would be helpful to show parasite stages on days -3 to -1 and indicate which day NAG is added. There is also some lack of consistency regarding usage of day 0/day 1: on line 911 (methods) the authors say that day 0 cultures are gametocyte rings, however on line 926 they also say gametocyte rings are included in the day 1 time point. Finally, previous work using the gametocyte promoter PEG4 (Brancucci et al. PMID: 26134953) suggests that expression of the reporter is not detectable in gametocyte rings--can the authors explain this discrepancy/their ability to use this marker for detection of committed rings/trophozoites?

How did cell composition vary across the two replicates and how were the datasets integrated? Please include a UMAP coloured by replicate or stacked bar chart that displays cells per cluster per replicate.

Supplementary figure 1F satisfactorily addresses this comment.

There is one cluster (C5) associated with males and it looks disconnected from the early gametocyte clusters in the UMAP and PCA. I wonder if this is a numbers/sex ratio issue. If you

downsample your female lineage to have equal representation of both sexes, how does that change your clustering and trajectory analysis?

The additional analysis performed in response to this comment satisfactorily addresses the reviewer's concerns and, as the authors note, provides a nice validation of their findings. I am inclined to agree with their assessment that this analysis does not need to be included in the manuscript given that down-sampling did not significantly affect the clustering or trajectory analysis.

Sup Fig 2B: Based on your comparison with the Biljon dataset, it seems like you have mostly later stage gametocytes or that your trajectories don't match real time changes. Could you please elaborate on this in the text?

The comparison of the correlation plots between the bulk and single cell data 266-273 is not very informative. I would remove and replace with a more detailed analysis of how your clusters match cells in real time.

The additional analysis performed by the authors comparing their data to the van Biljon bulk gametocyte transcriptome is informative, despite the challenges rightly point out by the authors in comparing single cell and bulk sequencing data. I would recommend replacing the previous Supplementary Figure 2 with Rev Figure S3, as proposed by the authors. For clarity, please be sure to clearly distinguish in the text when referring to cell type labels (stage and time point identifiers) from the van Biljon bulk dataset from this study's designation of days.

Additionally, you could strengthen your staging by integrating with the single-cell data from Gomes et al 2023 that has both the WT and md-1 KO data.

Again, the additional analysis performed by the authors is informative and nicely complements their findings. I would recommend incorporating these data into additional supplementary figure, space permitting.

Figure 5: Generally the figure and the associated text was difficult to digest. Could you expand on how this analysis relates to the genes that came up with the highest velocities in Figure 4, wouldn't those also qualify as driver genes? I think a clearer definition, both written and statistical, of driver gene is needed.

The added clarification of how driver genes are defined and explanation of how these differ from the highest velocity genes is useful and addresses the question raised by the previous Reviewer 2.

How does your model allow for the inference of upstream factors in sex determination? Perhaps using md-1 as your control could allow you to validate the model. How did you select the genes for a more detailed analysis in D and E?

The explanation of the selection criteria for genes used for further analysis (3' exon-intron junction coverage) is clear.

Other comments:

All responses to minor comments raised by the previous Reviewer 2 are satisfactory.

All responses to reviewer comments are addressed below in **RED**, line numbers refer to the new “marked up” version of the manuscript.

REVIEWERS' COMMENTS

Reviewer #1 (Remarks to the Author):

The authors appear to have addressed all of my and the other reviewers concerns. This remains an important piece of work and I look forward to seeing it in press.

We thank reviewer #1 for the previous feedback, which we believe has improved the quality of the manuscript significantly.

Reviewer #4 (Prev. Reviewer #2)

Single-cell transcriptomics reveal transcriptional programs underlying male and female cell fate during *Plasmodium falciparum* gametocytogenesis (revised manuscript)

In this study by Mohammed et al., the authors performed single-cell RNA seq on *P. falciparum* gametocytes to further characterise mechanisms of sex determination. They identify different transcriptional programs associated with male and female developmental, as well as putative driver genes for the male and female trajectories. Additionally, the authors perform motif-driven gene regulatory network analysis that allows them to predict how sex discrimination is regulated.

As described by the previous Reviewer 2, this is a large and very comprehensively analysed data

set that significantly contributes to our understanding of male and female gametocyte development in *P. falciparum*. The revised manuscript is much improved; however, some issues raised by the previous Review 2 remain outstanding.

Major comments:

Generally, the text could be improved from removing redundancies and being more concise. I found there was a lot of explanation of pretty standard analysis methods in both the results and the methods.

The text remains very long and would benefit significantly from additional trimming/editing for conciseness. The discussion section in particular repeats many of findings described in the results

and should be more tightly focused on analysing the implications of the data.

We thank the reviewer for these suggestions and have significantly trimmed the text throughout the results and discussion sections, including sections indicated at the following lines: 190, 268, 451, 453, 490, 558, 559, 647, 888, 1023, 1107, 1113, 1123, 1140, 1169, 1187, 1413,

More detail is needed on the induction method and corresponding time points both in Figure 1 and results as well as the methods. The authors reference the Fivelman paper but don't specify their exact induction conditions. For Figure 1, it would be helpful to have more timing information displayed on 1A. Also, illustrate that the four tubes were pooled and that you have two replicates.

Some additional information would still be warranted here. It would be helpful to indicate somewhere in the methods that multiple cultures were maintained in parallel so that all stages

could be collected on the same day (if this was the case). In the figure (panel 1A), the shaded arrow with FACS written underneath is confusing and suggests samples are collected on days -3 to -1. To further clarify induction method, it would be helpful to show parasite stages on days -3 to -1 and indicate which day NAG is added. There is also some lack of consistency regarding usage of day 0/day 1: on line 911 (methods) the authors say that day 0 cultures are gametocyte rings, however on line 926 they also say gametocyte rings are included in the day 1 time point. Finally, previous work using the gametocyte promoter PEG4 (Brancucci et al. PMID: 26134953) suggests that expression of the reporter is not detectable in gametocyte rings--can the authors explain this discrepancy/their ability to use this marker for detection of committed rings/trophozoites?

We thank the reviewer for highlighting this section. We have now changed panel 1A in Figure 1, and added information on lines: 1485-1490, in order to further clarify the parts relating to cell culture and FACS enrichment in the Methods section.

Note that we refer to developmental stages in accordance with Fivelman et al, where days -2 and -1 refer to the commitment cycle, and where day 0 is the first day of gametocyte development (after egress and reinvasion) followed by day 1 and so on. We therefore argue that this part in the Methods section now provides enough information.

We do not claim that rings (day -2) or trophozoites (late day -2 / early day -1) express fluorescence. What we claim is that our FACS enriched gametocyte populations at the different days include a somewhat heterogeneous population since the gametocytes do not develop perfectly in synchrony. Encapsulation of multiple single gametocytes from each enriched time point will therefore represent some level of developmental heterogeneity stretching several hours in either direction of the particular time point that was sorted. Therefore, we anticipate that our FACS enrichment at day 1 includes gametocytes of a several-hour timespan, including gametocytes representing the "day 0" developmental stage and include the earliest gametocyte developmental stage, i.e. gametocyte rings.

How did cell composition vary across the two replicates and how were the datasets integrated? Please include a UMAP coloured by replicate or stacked bar chart that displays cells per cluster per replicate.

Supplementary figure 1F satisfactorily addresses this comment.

Done - we thank the reviewer for the input.

There is one cluster (C5) associated with males and it looks disconnected from the early gametocyte clusters in the UMAP and PCA. I wonder if this is a numbers/sex ratio issue. If you downsample your female lineage to have equal representation of both sexes, how does that change your clustering and trajectory analysis?

The additional analysis performed in response to this comment satisfactorily addresses the reviewer's concerns and, as the authors note, provides a nice validation of their findings. I am inclined to agree with their assessment that this analysis does not need to be included in the manuscript given that down-sampling did not significantly affect the clustering or trajectory analysis.

Done - we thank the reviewer for the input.

Sup Fig 2B: Based on your comparison with the Biljon dataset, it seems like you have mostly later stage gametocytes or that your trajectories don't match real time changes. Could you please elaborate on this in the text?

The comparison of the correlation plots between the bulk and single cell data 266-273 is not very informative. I would remove and replace with a more detailed analysis of how your clusters match cells in real time.

The additional analysis performed by the authors comparing their data to the van Biljon bulk gametocyte transcriptome is informative, despite the challenges rightly point out by the authors in

comparing single cell and bulk sequencing data. I would recommend replacing the previous Supplementary Figure 2 with Rev Figure S3, as proposed by the authors. For clarity, please be sure to clearly distinguish in the text when referring to cell type labels (stage and time point identifiers) from the van Biljon bulk dataset from this study's designation of days.

We thank the reviewer for this suggestion and have replaced Supplementary Figure 2 with the suggested Rev Figure S3 and have modified the text accordingly in legends as well as in the manuscript on lines: 268-278.

Additionally, you could strengthen your staging by integrating with the single-cell data from Gomes

et al 2023 that has both the WT and md-1 KO data. (Rev Figure S2)

Again, the additional analysis performed by the authors is informative and nicely complements their findings. I would recommend incorporating these data into additional supplementary figure, space permitting.

We have added this as new supplementary figure (Supplementary figure 3), and added information to the Supplementary figure legend and in the text on lines 279-290

Figure 5: Generally the figure and the associated text was difficult to digest. Could you expand on

how this analysis relates to the genes that came up with the highest velocities in Figure 4, wouldn't

those also qualify as driver genes? I think a clearer definition, both written and statistical, of driver gene is needed.

The added clarification of how driver genes are defined and explanation of how these differ from the highest velocity genes is useful and addresses the question raised by the previous Reviewer 2.

Done - we thank the reviewer for the input.

How does your model allow for the inference of upstream factors in sex determination? Perhaps using md-1 as your control could allow you to validate the model. How did you select the genes for a more detailed analysis in D and E?

The explanation of the selection criteria for genes used for further analysis (3' exon-intron junction coverage) is clear.

Done - we thank the reviewer for the input.

Other comments:

All responses to minor comments raised by the previous Reviewer 2 are satisfactory.

Done - we thank the reviewer for the input.

Reviewer #3 (Remarks to the Author):

The authors have added two experiments to further evaluate the expression profile of the driver genes they identified and in the response to the reviewers demonstrated the concordance of their data with Gomes et al. The new data is a valuable addition and confirmed the predicted expression profile of the female driver genes and some of the male driver genes. The new data also clearly demonstrates the discovery nature of the data set. It is a useful tool toward dissecting male and female gametocyte development. However, as they state in the final paragraph further work is needed to experimentally confirm the functional and regulatory roles of the highlighted genes and lineages. The complicated pattern of the clusters designated early gametocytes could also benefit from additional focused evaluation of these stages. Consequently, throughout the manuscript terms like validated, regulate and mechanism need to be replaced or qualified. I have attempted to identify the sentences that need to be revised, but might have missed some that also need to be changed.

We thank the reviewer for their thorough feedback. As indicated by the reviewer, some clarifications needed to be addressed with regards to describing the findings and appropriate extrapolations. We have addressed the reviewer's comments regarding qualifications of our results as well as rephrased other characterizing words where appropriate.

Line 136 Clarify that the IMC is also produced in males (as discussed in line 431) and females, although differences were found in the expression of these associated genes.

This has now been clarified on lines 141 and 143:

Line 178 Please revise or remove the end of the sentence. The appearance of ETRAMP RNA in clusters C0, C3, C4 and C9 says nothing about protein transport, it just means ETRAMP RNA is expressed in cells present in clusters that are proposed to be early gametocytes.

The protein transport section has been removed, marked on line 190

213-225 Most of the gene listed here are not shown in Fig 2E. What are the profiles of the genes mentioned in the text, especially those that are mentioned as previously identified with early gametocytes. EXP1 (PF3D7_1121600), P113 (PF3D7_1420700), ETRAMP4 (PF3D7_0423700) and HDP1 (PF3D7_1466200)

The top genes in Fig 2E are not annotated and thus were not described. The genes highlighted were among the top 50 differentially expressed genes (Supplementary data 3) and have been implicated in early gam stages based on previous publications. Brief descriptions of their known functions have been added to this section on lines 238-241.

233 (Supplementary figure 1D, bottom panel). Change "validate" to "support". The results are from in silico analysis.

This has been changed

382-384 This sentence needs to be removed or extensively revised. No regulatory function has been shown for any of these genes or their splicing. The findings may suggest something that could be further tested, but by themselves they do not add significantly to our knowledge about the regulatory activity of transcription.

This sentence has been removed, i.e. *“These findings add an additional dimension to gene regulation during gametocytogenesis, including the regulatory activity of transcription based on spliced and unspliced mRNA”*

384-387 The different patterns of unspliced RNA in males and females are consistent with Gomes et al.'s finding but they do not directly support their data on the role of md1 splicing. To support their findings, you would have had to demonstrate differential splicing of md1 in your male and female populations, however this was not assessed and this sentence needs to be revised

This sentence has been removed, i.e. *“Furthermore, our findings are supported by recent data from Gomes et al.¹⁸, where they explored a series of splicing patterns of md1 gene regulation and matched them to the changes in male differentiation and development.”*

422 I think “recover” should be replaced with “reveal”

This has been replaced

431 Please clarify how this relates to the low RNA levels for many of the IMC associated genes shown in Figure 1G&H.

We believe that the reviewer is referring to Figure 2 G and H in light of lineage-based expression of IMC genes. Figures 2 F-H include differential expression analysis of the listed genes along the pseudotime axis, thus including differential gene expression analysis between early gametocytes and developing male and female gametocytes (i.e. there is no outlier control included in the analysis). Our analysis only highlights the relative difference in expression between the different stages/lineages included in our sample set and does not indicate an absence of expression among male gametocytes. Instead, it shows that there is a significantly higher abundance of IMC transcripts among females (and early developing gametocytes) as compared to developing male gametocytes.

We added “with *relatively* low to no expression of IMC genes within the male gametocyte lineage” on line 396

441-446 This section should be moved to the discussion and lines 441-443 tempered to reflect that it is all based on RNA sequencing. No actual experiments have been done evaluating the regulation of gene activity programs or the role of splicing kinetics in influencing sex transcriptional changes. This requires experimental modulation of the parameter to demonstrate the effect.

Lines 447-452 have been deleted from the results section.

453-455 Please revise this sentence. The RT-qPCR data used to confirm the expression of the

putative driver genes does not clearly demonstrate that there is lower expression at early time points for the 9 male genes, MD5 was not detected and MD4 & MD8 decline. From the individual cell sequencing presented later, there is also little difference between the expression of the driver genes and the two time points (early to mid and mid to late) and one of the male drivers is also expressed in some female gametocyte.

This sentence has been revised on lines 602-603. We previously addressed the lack of MD5 detection due to technical issues.

In Supplementary fig S5B&C The results from the individual genes at the 3 time points should be clustered together, not separated by time point, to allow direct analysis of the trend for each gene over time,

We have changed the axis to reflect individual genes with time points clustered in this figure (now Supplementary figure 6).

469 the same gene is listed twice.

The second gene has been replaced with the appropriate female marker, PF3D7_1031000

495 Does the analysis in Supplementary figure S6 use all the 9 male driver genes even those that did not work well in the RT-qPCR analysis?

(Supplementary figure 6 - now Supplementary figure 7). We included the top 20 hits of both male and female markers from the differential expression analyses found in supplementary data file 3, and have now added this information in a separate table (Supplementary table 3), as well as to the Methods section on lines 1844-1845

514 Please remove or revise the end of the sentence. No evidence is provided that these genes have a functional role in the sexual determination mechanisms. Some of them could be markers to define the lineage, but this does not imply a regulatory relationship. None of the driver genes in Table 6 have any annotated or experimental role in gene regulation.

The sentence, "The functional divergence of the putative male and female driver genes..." has been removed.

532-541 Move this to the discussion.

This section has been removed

600-602 Move this to the discussion.

This section has been moved to line 1271-1274.

603 Add " is predicted to" before regulates

This has been added

628 Add "predicted" before "repressor"

This has been added

648 “ensure validation” is too strong please replace with the following or similar: We used previously reported canonical markers of mature gametocytes to confirm the mapping of differentiated mature male and female gametocyte cell populations in our single-cell dataset

This has been replaced

655 add “selected” before “putative driver genes”. Not all 100 driver genes were tested.

This has been added

675 Please rephrase this. The results confirm an increase in expression over time for the female genes tested, but not the male genes (Supplementary Fig 5A,B&E).

This section has been removed

680 This needs to be qualified in that a number of male driver genes were not consistently or selectively expressed in the males Supplementary Fig 5E. There was also no difference in gene expression between individually captured early-mid and mid-late gametocytes in Supplementary Fig 5E.

This is described in the results section, lines: 602-603nd in the discussion, line 1085

688-692 Both sentences should be removed. No evidence is presented for MD7 to be essential in males, not females. Nor is any data provided that most lineage markers are essential for development. The profile for MD7 is quite different from either MM1 or MM2. I assume MM2 is PF3D7_1438800, md1 of Gomes et al..

These two sentences have been removed

The definition of the MMs and FMs should be given in the figure legend for Supplementary Fig 5.

These definitions have been added in the supplementary figure legends (now supplementary figure 6)

705-706 Please remove this phrase. The male and female driver genes have a dN/dS significantly above nuclear genes and no other evidence is presented supporting the statement that they are essential for parasite fitness and survival.

This phrase has been deleted

709-726 Could be removed or significantly shortened. The differential expression of a gene in males and females is not a new insight into the potential role of TRF1 in gametogenesis.

This entire section has been removed, marked on line: 1140

755-761 It should be noted that the complex mix of clusters designated as early gametocytes

suggests that more work is needed to define the early steps of male and female gametocyte production.

The following sentence has been added to this section “However, comparisons between single-cell data from this paper and van Biljon et al support that further work is necessary to characterize the early gametocyte clusters associated with male and female production” on lines: 1226-1227

769 Change “validates” to “supports”

This has been changed

791 Add “that its expression profile suggests it” before “regulates at least two genes, of which one (PF3D7_0209500) has not yet been functionally annotated” No evidence is presented that AP2-G2 actually regulates these two genes.

This has been changed as suggested

826 Please revise this sentence to clarify its meaning.

This sentence has been clarified. “We then used the mGRN dataset to infer regulons, i.e. groups of genes regulated by a common regulatory protein, enriched in our single-cell dataset”, on lines: 1314-1315

831 Replace “validate” with “identify”. Validation implies confirming a functional link which was not done.

This has been changed

841 Change “indicating” to “suggesting”

This has been changed

860-878 This section is unrelated to the work reported in this study and should be deleted. Their study examines sexual development, not sexual conversion or epigenetic regulation. Sexual conversion is initiated during the asexual cycle, which disrupted by the application of NAG.

This section has been deleted

881 add “suggests” before “gene regulatory” and replace “mechanisms” with pathways.

These have been replaced

Figure legends: Please define the number of cells used as the denominator for the prevalence of the genes listed in the all the dot plots.

This information has been updated accordingly for the legends of main figures 1 and 2, as well as supplementary figures 1 and 8